# Retinitis pigmentosa–associated mutations in mouse Prpf8 cause misexpression of circRNAs and degeneration of cerebellar granule cells

Michaela Krausová[1], Michaela Kreplová[1], Poulami Banik[1] , Zuzana Cvačková[1], Jan Kubovčiak[1], Martin Modrák[2] , Dagmar Zudová[3], Jiří Lindovský[3] , Agnieszka Kubik-Zahorodna[3], Marcela Pálková[3], Michal Kolář[1] , Jan Procházka[3], Radislav Sedláček[1,3], David Staněk[1] 

**A subset of patients with retinitis pigmentosa (RP) carry mutations in several spliceosomal components including the PRPF8 protein. Here, we established two alleles of murine *Prpf8* that genocopy or mimic aberrant PRPF8 found in RP patients—the substitution p.Tyr2334Asn and an extended protein variant p.Glu2331ValfsX15. Homozygous mice expressing the aberrant Prpf8 variants developed within the first 2 mo progressive atrophy of the cerebellum because of extensive granule cell loss, whereas other cerebellar cells remained unaffected. We further show that a subset of circRNAs were deregulated in the cerebellum of both Prpf8-RP mouse strains. To identify potential risk factors that sensitize the cerebellum for Prpf8 mutations, we monitored the expression of several splicing proteins during the first 8 wk. We observed down-regulation of all selected splicing proteins in the WT cerebellum, which coincided with neurodegeneration onset. The decrease in splicing protein expression was further pronounced in mouse strains expressing mutated Prpf8. Collectively, we propose a model where physiological reduction in spliceosomal components during postnatal tissue maturation sensitizes cells to the expression of aberrant Prpf8 and the subsequent deregulation of circRNAs triggers neuronal death.**

## Introduction

The PRPF8 protein is the central scaffolding component of the spliceosome, which organizes its catalytic RNA core and directly regulates the activity of the key spliceosomal RNA helicase SNRNP200 (also known as Brr2) to control correct timing of spliceosome activation (Mozaffari-Jovin et al, 2014). Congenital mutations in genes encoding core spliceosome constituents including *PRPF8* (McKie et al, 2001), *SNRNP200* (Zhao et al, 2009), *PRPF3*

(Chakarova et al, 2002), *PRPF4* (Chen et al, 2014), *PRPF6* (Tanackovic et al, 2011a), and *PRPF31* (Vithana et al, 2001) have been identified as causative in a subset of familial blindness disorders known as non-syndromic retinitis pigmentosa (RP) (for a review, see Krausova & Stanek, 2018). These dystrophies of retinal tissues feature dysfunction of retinal pigment epithelium (RPE) and gradual loss of both photoreceptor types. The pre-mRNA splicing factor genes are systemically expressed; the narrow phenotypic restriction observed with the human RP disease, however, indicates enhanced sensitivity of certain tissues toward the expression of aberrant splicing factors.

Recent years have provided the first molecular insights into the pathophysiology of splicing factor forms of RP. The most studied is *PRPF31*, which is compromised by a spectrum of inactivating mutations (Vithana et al, 2003; Sullivan et al, 2006; Rio Frio et al, 2008; Huranova et al, 2009; Audo et al, 2010). RP mutations in *PRPF31* evoke mis-splicing of numerous retina-specific genes including rhodopsin and ciliary and cellular adhesion genes, which disrupt the structural organization of RPE and photoreceptors (Yuan et al, 2005; Mordes et al, 2007; Ray et al, 2010; Yin et al, 2011; Buskin et al, 2018). The etiology of *PRPF8*-linked RP comprises nearly two dozen heterozygous genetic variants (McKie et al, 2001; De Erkenez et al, 2002; Ruzickova & Stanek, 2017) with predominance of missense mutations clustering to very C-terminus of PRPF8 that is responsible for the SNRNP200 modulation (Mozaffari-Jovin et al, 2013). Functional screens demonstrated that most of the aberrant PRPF8 proteins exhibited altered ability to interact with spliceosome co-factors in human cell culture and impacted development including eye development in fly (Malinova et al, 2017; Stankovic et al, 2020). The only exception was the PRPF8 p.Tyr2334Asn substitution. This variant displayed in in vitro assays selective disruption of splicing efficiency and affected strongly pupation in *Drosophila*, but in contrast to the other pathological PRPF8 variants, the PRPF8 Y2334N protein was properly incorporated into small nuclear ribonucleoprotein particles (snRNPs), key components of the spliceosome

[1]Institute of Molecular Genetics, Czech Academy of Sciences, Prague, Czech Republic   [2]Core Facility Bioinformatics, Institute of Microbiology of the Czech Academy of Sciences, Prague, Czech Republic   [3]Czech Centre for Phenogenomics, Institute of Molecular Genetics, Czech Academy of Sciences, Vestec, Czech Republic

Correspondence: stanek@img.cas.cz

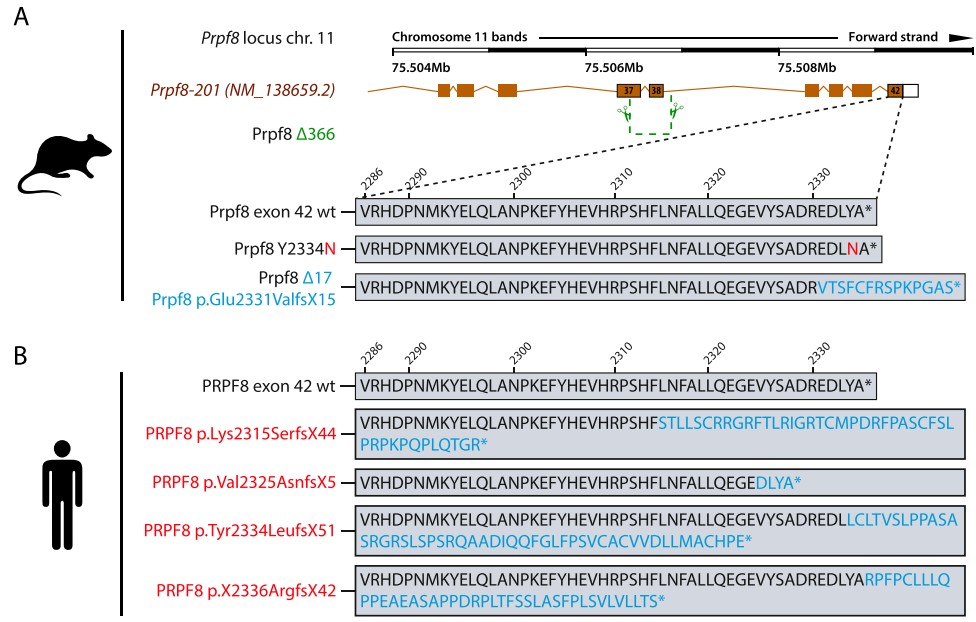

**Figure 1. Newly established murine Prpf8 alleles serve as a model for human RP-linked mutations.**
**(A)** Illustration of the three newly established *Prpf8* alleles: the RP-causative substitution Prpf8 p.Tyr2334Asn (in red letters); the *Prpf8^Δ17* allele with an aberrantly prolonged Prpf8 ORF (in blue); and the *Prpf8^Δ366* allele, where a large deletion of 366 bp eliminated the 3′ portion of *Prpf8* exon 37 until 5′ of introns 38 and 39 (in green). **(B)** *Prpf8^Δ17* allele serves as a genetic model for human RP-linked mutagenic frameshifts that map to the PRPF8 distal region (in red). See Table S1 and Fig S1 for primer design and sequencing of the founders and Fig S2 for analysis of mutated offspring.

(Malinova et al, 2017; Stankovic et al, 2020). In addition, several RP-linked mutations in *PRPF8* target codons at the very 3′ end of the PRPF8-coding sequence causing shortening or aberrant prolongation of the PRPF8 C-terminus (McKie et al, 2001; De Erkenez et al, 2002; Martinez-Gimeno et al, 2003; Tiwari et al, 2016).

Several of the RP-causing PRPF8 missense mutations have been recapitulated in model organisms. Experimental animals homozygous for the aberrant substitutions Prpf8 p.His2309Pro (Graziotto et al, 2011) or p.Arg2310Gly (Kukhtar et al, 2020) were viable, ruling out a full Prpf8 loss-of-function effect because full genetic deficiency in *Prpf8* was embryonic lethal (Graziotto et al, 2011). However, although challenged to homozygosity the *Prpf8^H2309P/H2309P* mice manifested mild, very late-onset retinal degeneration (Graziotto et al, 2011; Farkas et al, 2014). We are thus still lacking a robust vertebrate model that would clarify the disease-provoking mechanisms associated with distal C-tail–disrupted Prpf8 variants and that would, moreover, elucidate principles underlying sensitivity of specific cell types to *PRPF8* RP mutations.

Here, we genocopied the PRPF8 p.Tyr2334Asn missense mutation in a mouse because the PRPF8 p.Tyr2334Asn mutation confers severe clinical phenotype with early macular involvement and the molecular mechanism is unclear (Towns et al, 2010). In addition, we also generated an extended Prpf8 protein variant (*Prpf8^Δ17*) that resembles human mutagenic frameshifts originating from PRPF8 C-terminal residues and a *Prpf8* knock-out allele that serves a direct comparison with a presumptive *Prpf8* loss-of-function mode.

## Results

### Generation of the *Prpf8^Y2334N* and *Prpf8^Δ17* mice

To introduce the Prpf8 p.Tyr2334Asn mutation to the mouse C57BL/6N background, we supplemented CRISPR/Cas9 gene editing tools with DNA-based homology-directed repair (HDR) templates (Fig S1A and Table S1). In parallel to the correctly established *Prpf8^Y2334N* allele in two founder animals, in a third individual the non-homologous end-joining repair pathway gave rise to a *Prpf8* deletion allele, where removal of terminal 17 base pairs (bp; nucleotides chr11:75,509,271–11:75,509,287; hereafter termed the *Prpf8^Δ17* allele) within the Prpf8 open reading frame abolished five C-terminal amino acids including the stop codon. This allele thus encodes a Prpf8 protein variant with altered residues aa2,331–2,335 that is moreover extended by aberrant nine amino acids at the C-terminus (Prpf8 p.Glu2331ValfsX15; called Prpf8^Δ17 in the following text; Fig 1A). The *Prpf8^Δ17* allele suitably resembles human PRPF8 RP variants where mutagenic frameshifts originate from the stop codon itself, or from mutations within or shortly upstream of the penultimate residue Tyr2334, and we therefore included this variant in further analysis (Martinez-Gimeno et al, 2003; Tiwari et al, 2016) (Fig 1B). A detailed sequencing analysis of the targeted *Prpf8* regions in all three founder animals is shown in Fig S1B.

In previous in vitro experiments in human cells, the PRPF8 Y2334N substitution did not alter the binding of PRPF8 with its key interactors (Malinova et al, 2017); the effect of the Prpf8^Δ17 mutation is, however, unclear. To inspect the binding profile of the aberrant Prpf8 variants, we performed immunoprecipitation of Prpf8 from cerebellar tissues and monitored co-precipitation of dedicated Prpf8 binding partners Prpf6 and Snrnp200 (Fig S1C). Both mutated Prpf8 variants pulled down similar amounts of Prpf6 and Snrnp200 when compared to the wt protein, which indicated that neither of the mutations perturbed the Prpf8 interactions with these two U5 snRNP-specific factors.

All novel *Prpf8* strains were generated on the default C57BL/6N genetic background given the higher efficiency of editing efforts we achieved with C57BL/6N-derived zygotes. However, the C57BL/6N lines carry the Crumbs homolog 1 (*Crb1*) *rd8* mutation that is known to progressively compromise photoreceptor integrity (Mattapallil

et al, 2012). To avoid any effect of the *rd8* mutation, we bred the founders to the C57BL/6J genetic background that does not carry the mutation at the *Crb1* locus. For phenotypic analyses, progeny was successively back-crossed to C57BL/6J for >7 generations when the contribution of the recurrent parent genome reaches >99% (Visscher, 1999). This approach also eliminated potential off-target effects of CRISPR/Cas9 editing.

## Homozygous *Prpf8^{Y2334N/Y2334N}* and *Prpf8^{Δ17/Δ17}* mice develop cerebellar neurodegeneration

Both newly established *Prpf8^{Y2334N}* and *Prpf8^{Δ17}* lines were cross-bred to homozygosity. Pups were viable and born in Mendelian ratios, but we recorded a mild shortage of *Prpf8^{Δ17/Δ17}* offspring and a surplus of *Prpf8^{Y2334N/Y2334N}* progeny (Fig S2A and B) at the expense of heterozygous animals. Homozygous animals of both strains were moreover slightly smaller compared with their heterozygous and wt littermates of matched gender. In *Prpf8^{Δ17/Δ17}* mice, the difference in body weight was already noticeable at 4 wk (lower by 13% and 12% in males and females, respectively) and persisted to later ages (15% reduction in males and 12% in females at 8 wk) (Fig S2C). The *Prpf8^{Y2334N/Y2334N}* animals showed a smaller weight disproportion (6% decline in males and 8% in females at 12 wk of age) (Fig S2C).

When reaching ~15 wk of age, homozygous *Prpf8^{Y2334N/Y2334N}* and *Prpf8^{Δ17/Δ17}* animals exhibited tremor, indicating potential neurodegeneration. To investigate the onset and progression of the suspected neurodegenerative changes, we histologically examined cerebellar structures at 4, 6, 15/17, and 22 wk of age (Fig S3 for *Prpf8^{Y2334N/Y2334N}* animals and Fig S4 for *Prpf8^{Δ17/Δ17}* mice). The postnatal maturation of the cerebellar tissue was normally completed; however, at week 6, we observed a reduction in cellularity in the granule cell layer in the posterior lobe (Figs S3A and B and S4A and B). In aging mice, progressive thinning in the stratum granulosum roughly followed the anatomical subdivision of the cerebellar tissue, and overall, hemispheric segments of lobules seemed more strongly afflicted compared with vermian regions (Fig S3C). The granular layer content was severely decreased by 17 and 15 wk of age in *Prpf8^{Y2334N/Y2334N}* and *Prpf8^{Δ17/Δ17}* animals, respectively (Figs S3D and S4C), which coincided with the onset of tremor and locomotion disturbances (see Video 1). By 22 wk of age, the granule cell layer almost diminished in the whole cerebellum (Figs S3E and S4D), and mice had to be euthanized because of extensive ataxia. Heterozygous peers of *Prpf8^{Y2334N/wt}* and *Prpf8^{Δ17/wt}* genotypes did not develop any gross pathological changes by the ultimate timepoint examined (Figs S3E and S4D).

In the latest 22-wk cohorts of both Prpf8 lines, we histologically inspected specimens from spinal cord, peripheral nerves, skeletal muscle, and eye, but did not observe any pathological changes (Fig S5). In the aged *Prpf8^{Δ17/Δ17}* animals, we also examined the extracerebellar populations of granule cells that reside in the olfactory bulb and dentate gyrus, but no apparent perturbances were observed (Fig S6A–C). To survey the condition of retinal structures, both eyes of the *Prpf8^{Δ17}* 22-wk cohorts were screened by paralleled optical coherence tomography (OCT) and ERG examinations. The ophthalmic inspection of fundi did not reveal any abnormalities in retinal layering, nor in the size and placement of the head of the optic nerve, and the structure and distribution of superficial blood vessels as well did not vary from wt controls. The retinal layers were well developed; however, in homozygous *Prpf8^{Δ17/Δ17}* animals we recorded a slight, but significant, reduction in the retinal thickness (Fig S6D and E). In the electroretinographic examinations, all mice showed scotopic and photopic responses with normal waveform shape preserved. Despite large variability among animals, in homozygous mutants the responses were slightly smaller in amplitude and delayed in time when compared to *Prpf8^{wt/wt}* and *Prpf8^{Δ17/wt}* counterparts. This effect seemed to be more pronounced in wave b rather than in wave a, suggesting that the photoreceptor function was affected to a lesser extent than post-photoreceptor retinal processing (Fig S7). The subsequent histopathological inspection of ocular tissues nonetheless did not reveal any apparent retinopathy (Fig S6F). Taken together, the cerebellar cortex is likely the primary site perturbed by the presence of both aberrant Prpf8 variants.

**Prpf8 mutations induce changes in the cerebellar RNA landscape**

To investigate how the mutations in splicing factor Prpf8 affect the transcriptional landscape of the cerebellum, total RNA was isolated from cerebella of 12-wk-old mice of the *Prpf8^{Y2334N}* strain and from 8-wk-old *Prpf8^{Δ17}* animals, and subjected to next-generation RNA sequencing (RNA-Seq). These timepoints were selected to precede the manifestation of tremor and locomotion disturbances, and represented two consecutive stages of the granular layer degeneration. In more detail, at the earlier timepoint of 8 wk the *Prpf8^{Δ17/Δ17}* animals displayed apoptosis occurring in the granular layer in apical portions of the posterior lobe, whereas the vermian regions and the anterior cerebellum segment were less strongly afflicted (Fig 2A and C). The more advanced stage of the *Prpf8^{Y2334N/Y2334N}* mice was, in comparison, characterized by a noticeable reduction in granule cell density in the posterior apices that suggested preceding clearance of the deceased neurons and residual debris by phagocytic microglia (see below). Neuronal death has by then proceeded to lobule bases and was present in the anterior compartment (Fig 2B and D).

The RNA-Seq analysis revealed significant changes in gene expression, and hundreds of genes were up- and down-regulated in both Prpf8 mutant lines but not in the heterozygote animals (Fig 3A and Table S2). Most of the deregulated genes are protein-coding genes, and we identified only 20 lincRNAs that are deregulated in both homozygous strains. To explore the likely fate of individual cerebellar cell types, a set of classificatory marker genes was extracted from a cerebellar single-cell RNA-Seq profiling study (Saunders et al, 2018) as a proxy that delimited homeostatic cerebellar populations (Fig 3B). In concordance with the prior histopathological findings, we observed in homozygous animals a decline in the expression of well-established granule cell markers including *Gabra6*, *Rbfox3*/NeuN, and *Reln*. We also detected a preferential decline in genes physiologically enriched in the granular layer of the posterior cerebellum such as paired box 6 (*Pax6*), and T-cell leukemia homeobox 3 (*Tlx3*) (Divya et al, 2016) (Fig 3C), which was consistent with the observed onset of degeneration in the posterior region (Fig 2).

Concerning the synapse organization, we recorded a drop in prominent scaffolding proteins of the active zone cytomatrix

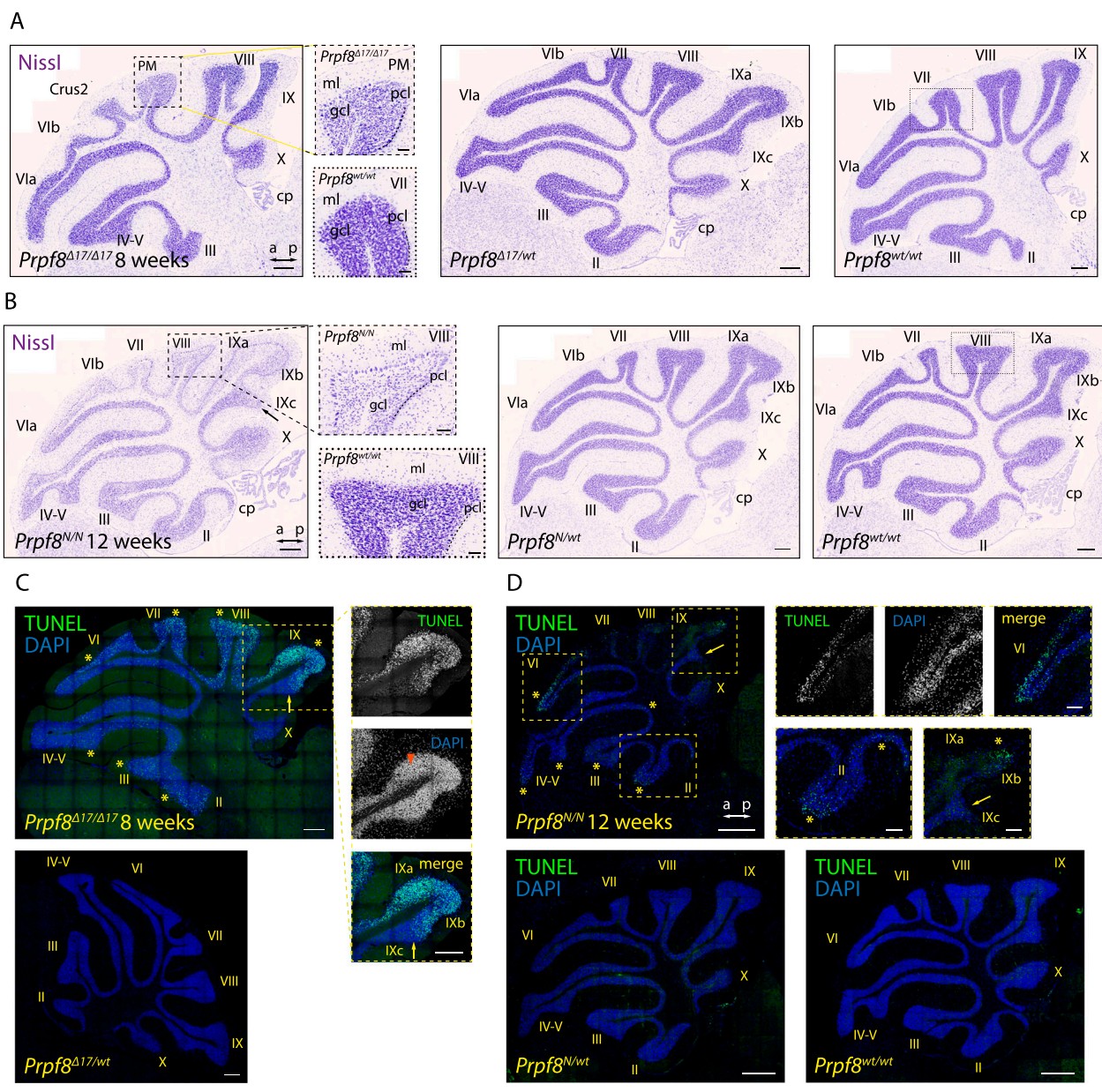

**Figure 2.  Aberrant Prpf8 variants provoke degeneration of the cerebellar granule cell layer.**
Histopathological analysis and TUNEL assay on cerebellar specimen collected from 8-wk-old *Prpf8*^Δ17^ animals and from 12-wk-old *Prpf8*^Y2334N^ mice. **(A, B)** Nissl staining on sagittal cerebellar sections acquired from *Prpf8*^Δ17^ (A) and *Prpf8*^Y2334N^ (B) individuals. **(A)** In 8-wk-old *Prpf8*^Δ17/Δ17^ mice, we detected shrunken granule cells in apical portions of the posterior lobe (the detail is from paramedian [PM] lobule). **(B)** In 12-wk-old homozygous *Prpf8*^Y2334N/Y2334N^ animals, we observed a substantial reduction in granule cell density in apices of the posterior lobe (the detail is from lobule VIII [dashed rectangle]). The loss of granule cells followed the anatomical subdivision of the cerebellum (note the sharp boundary between posterior lobule IXb and lobule IVc that belongs to the nodular lobe [black arrow]). a, anterior; cp, choroid plexus; Crus2, crus 2 of the ansiform lobule; gcl, granule cell layer; ml, molecular layer; p, posterior; pcl, Purkinje cell layer. Scale bar = 200 µm; in details, scale bar = 50 µm. For additional timepoints and tissues, see Figs S3–S7. **(C, D)** Visualization of apoptotic cells by TUNEL assay. **(C)** In 8-wk-old *Prpf8*^Δ17/Δ17^ cerebella, neuronal death was present in apices of the posterior lobe, with numerous shrunken granule cells detected (orange arrowhead). Note the strict apoptotic boundary between lobules IXb and IXc (yellow arrow). Scale bar = 200 µm. **(D)** In 12-wk-old *Prpf8*^Y2334N/Y2334N^ animals, apoptosis was prevalent in the apex of lobule VI and was also present in the granular layer in both apical and base regions of the anterior lobe (yellow asterisks). The apoptosis followed the anatomical division of the cerebellum, for now sparing granule cells in lobule IXc that belongs to the nodular lobe (yellow arrow). No apoptotic signal was recorded in heterozygous *Prpf8*^Y2334N/wt^ animals (bottom row). Scale bar = 200 µm; in insets, sale bar = 50 µm. The numbers of animals in individual cohorts are shown in the Materials and Methods section.

including *Bsn* and *Erc1*, and a decrease in multiple small GTPase effectors that play an essential role in neurotransmitter release (*Rims1*, *Rims2*, and *Rims3*) (Fig S8A and C). Homozygous mutant cerebella displayed also reduction of genes involved in the Wnt signaling cascade (*Wnt7a*, *Dvl1*, and *Fzd7*, but not *Fzd5*), which indicated perturbance of the Wnt7a signaling axis present at the dendritic, post-synaptic side of glomerular rosettes (Ahmad-Annuar et al, 2006). Altogether, the spectrum of the down-

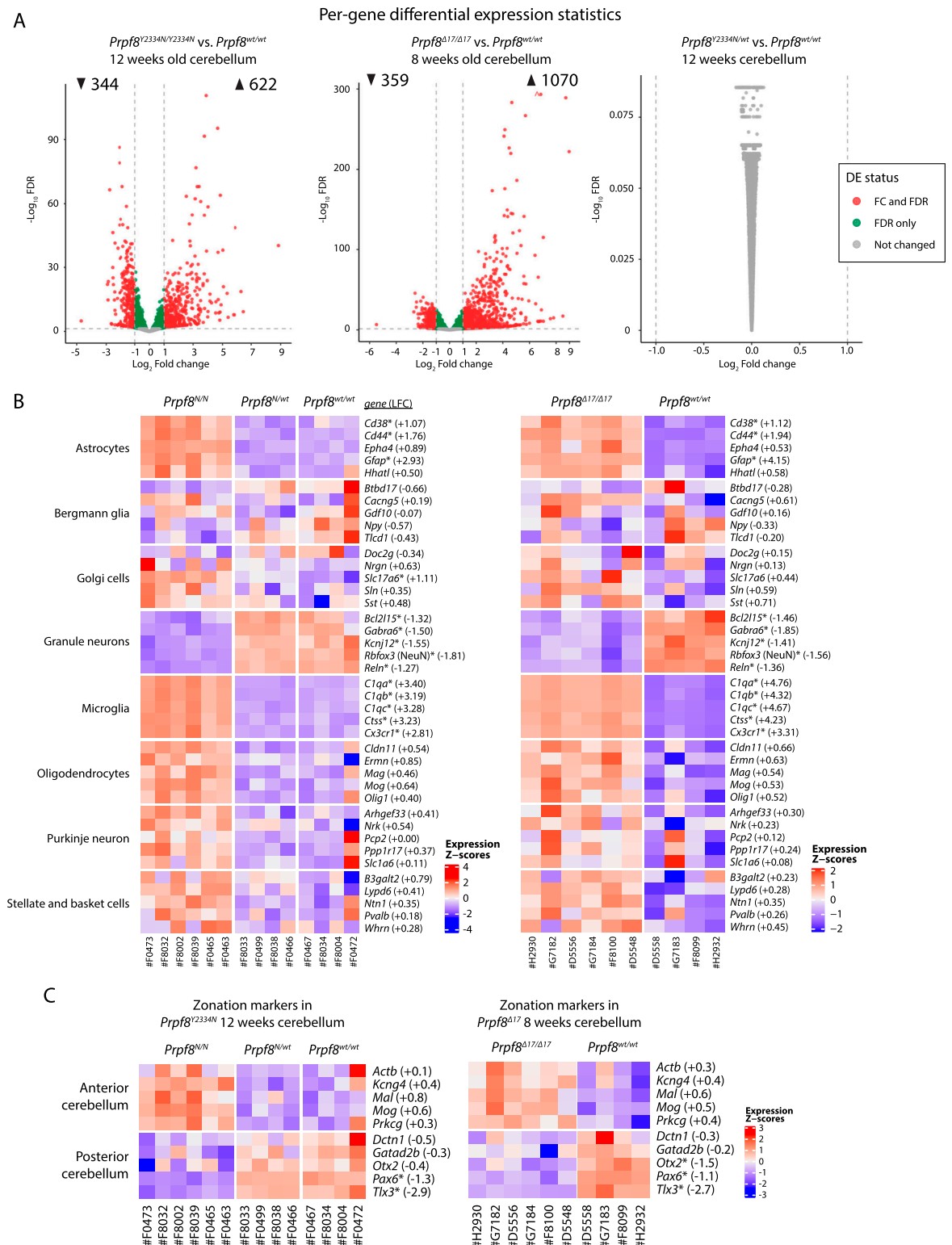

**Figure 3. Differential gene expression in Prpf8 mutant cerebella.**
RNA was isolated from cerebellar samples collected of 12-wk-old animals of the *Prpf8*[Y2334N] strain, or 8-wk-old *Prpf8*[Δ17/Δ17] animals and their *Prpf8*[wt/wt] littermates, and analyzed by RNA-Seq. **(A)** Volcano plot depicting –log$_{10}$ false discovery rate (FDR) versus log$_2$ fold change difference in gene expression between Prpf8 homozygous mutant mice and *Prpf8*[wt/wt] controls. Red circles denote significantly up- and down-regulated genes (FDR<0.05, fold change cutoff is set at 1; see also Table S2). n = 6 for *Prpf8*[Δ17/Δ17] and *Prpf8*[Y2334N/Y2334N] biological replicates and n = 4 for *Prpf8*[wt/wt] litter-matched controls (both strains). **(B, C)** Heatmap representation of gene expression levels of selected marker genes representing major cerebellar homeostatic cell types (B) and anterior/posterior parts of the cerebellum (C). Values were standardized to

regulated genes suggested loss of components specific to axonal and dendritic synaptic zones of granule cells.

The category of up-regulated genes was enriched with a multitude of microglial markers, which were previously identified in microglial populations associated with disease and neurodegeneration conditions (Paolicelli et al, 2022) (Figs 3B and S8B and C). We also observed a slight rise in transcripts characteristic of oligodendrocytes, such as *Mag* and *Mog*, which can originate in the physiological enrichment of myelination factors in the anterior cerebellum (Divya et al, 2016), and may reflect the disproportion of representation of specialized cerebellar cell types in the degenerating tissue. Similarly, the small increase in Purkinje neuron–specific genes might be linked to their natural overrepresentation in the anterior lobe and can reflect the regional variation in Purkinje cell populations within the cerebellum.

To confirm the expression profiling, we detected selected proteins using immunocytochemistry in cerebellar sections of 8- and 12-wk-old animals (Fig S9). In accordance with RNA-Seq results, we observed down-regulation of granule cell marker Rbfox3 (NeuN) and synaptic marker PSD95 (encoded by *Dlg4*) in wider pre-apoptotic areas, which might hallmark distressed neurons and precede the upcoming cell death (Fig S9A and B). Consistent with RNA-Seq data, the granule cell layer was also strongly positive for astrocytes (Fig S9C) and CD45⁺ microglia (Fig S9D). We did not observe any major alterations affecting Purkinje neurons and Bergmann glia (Fig S9E and F). The blend of histopathological and transcriptomic analyses revealed substantial changes occurring in the cerebellar tissue of homozygous Prpf8 mutant animals; namely, the degeneration of granule cells in the posterior lobe was accompanied by extensive induction of microglia and astrogliosis.

### Deregulation of circular RNA expression in the cerebellum of homozygous *Prpf8^{Y2334N/Y2334N}* and *Prpf8^{Δ17/Δ17}* mice

We further analyzed RNA-Seq data for changes in alternative splicing (Table S3), but given the distorted representation of cerebellar cell subtypes, it was not feasible to attribute the recorded differences to Prpf8 mutations only. The analysis of alternative splicing nonetheless revealed altered usage of exons, which were previously shown to be included in circular RNAs (Rybak-Wolf et al, 2015). A detailed analysis focused on the circRNA expression revealed that the cerebella of mutant mice exhibited considerable deregulation of the circular transcriptome (Fig 4 and Table S4). In detail, we observed 161 (*Prpf8^{Δ17/Δ17}*) and 26 (*Prpf8^{Y2334N/Y2334N}*) circRNAs more than twice elevated, and on the contrary, we recorded more than a twofold decline in 63 (*Prpf8^{Δ17/Δ17}*) and 34 (*Prpf8^{Y2334N/Y2334N}*) circRNAs (Fig 4A). Interestingly, in both strains most of the significantly deregulated circRNAs were altered by a larger factor compared with the shift in host gene abundance, and there were also classes of circRNAs that followed an inverse scenario compared with the change in their parental mRNA (Fig 4B). Several

abundant up- and down-regulated circRNAs were selected (Table S5) and changes in expression confirmed by RT–qPCR (Fig 4C).

The disequilibrium in the cerebellar circRNAs in the older mice might have been influenced by the altered representation of individual cell types in the degenerating tissue. To investigate whether cerebellar circRNAs are perturbed already before the onset of the granule cell decay, we examined the cerebellar transcriptome by RNA-Seq approach in 4-wk-old animals. At this timepoint, we did not detect any pathological changes by immunohistochemistry, TUNEL assay, and RT–qPCR approaches (Fig S10A–E). The RNA-Seq profiling revealed no substantial differences in the expression of linear RNA forms between wt and homozygous mice of either strain (Fig S11A), and we did not record any significant deregulation of circRNAs either (Fig S11B). However, a more detailed analysis of circRNA expression in the 4-wk-old animals by RT–qPCR revealed that the expression of several circRNAs was already shifted in the same direction as in the degenerated cerebellum. We observed a decline in circ_0015493 produced from *Ntm* and *Rims1* circ_0000017 and *Rims2* circ_0000595 derived from *Rims1* and *Rims2* synaptic genes, respectively (Figs 5A and B and S11C). *Rims1*- and *Rims2*-derived circRNAs are highly abundant in the cerebellum, and *Rims2* circ_0000595 represented the dominant form of total *Rims2* transcripts in 4-wk-old and older animals (Fig S11C and Table S4) (Rybak-Wolf et al, 2015). Importantly, we observed a significant drop in *Rims1* circ_0000017 amounts, whereas other cognate circRNAs formed from the same locus, such as *Rims1* circ_0000016 and *Rims1* circ_0000018, were changed by a lesser extent (Fig 5A and B). In addition, we observed a significant decline in *Ntm* circ_0015493 in the predegenerative cerebella of both Prpf8 mutant strains (Figs 5A and B and S11D) but not the cognate linear mRNA (Fig S12). Similar to *Rims2* circ_0000595, the *Ntm* circRNA entity represented the prevailing form of total cerebellar *Ntm* content (Table S4). We further analyze circRNA-specific exons that were only marginally included in linear mRNA (Ntm and Rims1) or in the case of the *Rims2* gene were fully circRNA-specific (Fig S11C and D and Table S6) (Rybak-Wolf et al, 2015). Both transcriptome-wide analysis and RT–qPCR in 4-wk-old animals revealed lower inclusion of these circRNA-specific exons, which further documented the lower expression of the cognate circRNAs (Table S6 and Fig 5A and B).

Analyses of splice site (ss) strength scores collectively identified a weak 3′-ss rating for *Rims1* alternative exons A and B, for the *Rims2* exon 22, and for the *Ntm* alternative exon B′′ (Table S7). These data indicate that the suboptimal 3′ ss that surround these alternative exons are likely not properly recognized by the mutated variants of Prpf8, and this finding might explain why only some circRNA species originating from the same host gene portion are selectively affected.

In contrast to circRNA down-regulation, several circRNAs (but not linear mRNAs) were up-regulated in the predegenerative stage (Figs 5A and B and S12), and this elevation persisted to later stages in both Prpf8 mutant strains (Fig 4C). In detail, we examined the expression of *Rbfox1* mmu_circ_0006379, *Zfp521* mmu_circ_0007146, and *Chd7*

display the same range of expression values for each gene. *, differentially expressed genes defined by |log₂ fold change|>1 and FDR<0.05; the estimated log₂ fold change (LFC) value is provided for individual genes and represents the difference between the homozygous mutant and *Prpf8^{wt/wt}* cohorts. The expression of selected marker genes was validated by RT–qPCR and is shown in Fig S8C.

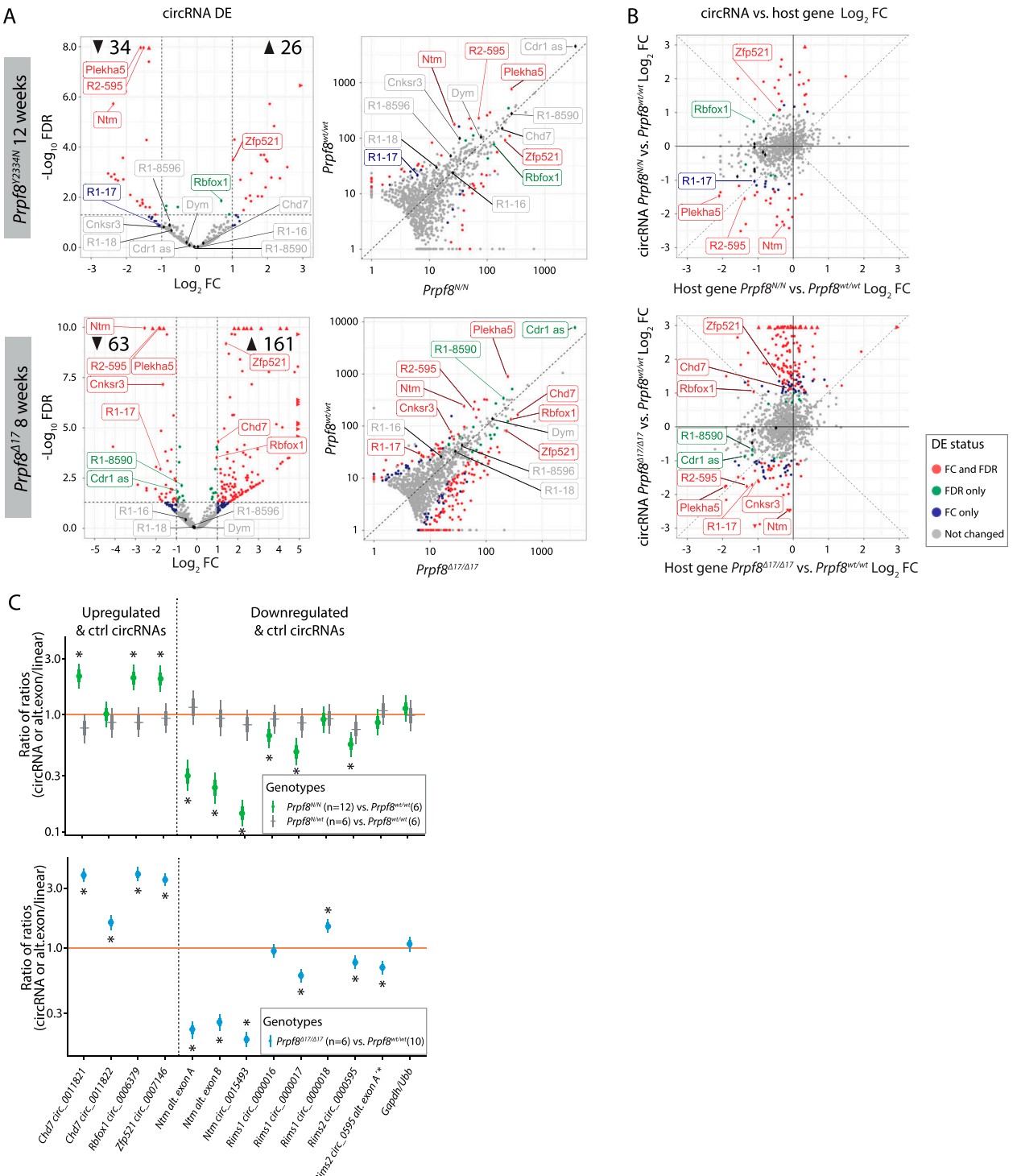

**Figure 4. Deregulation of circular RNAs in cerebellar tissue of homozygous Prpf8 mutant mice.**
**(A)** Volcano plot showing –$\log_{10}$ (FDR) versus $\log_2$ fold difference in circular RNA expression between 12-wk-old $Prpf8^{Y2334N/Y2334N}$ animals (top row), or 8-wk-old $Prpf8^{\Delta17/\Delta17}$ animals (second row) and their $Prpf8^{wt/wt}$ littermates. Scatter plot comparing library-normalized abundance of individual circRNAs (in BSJ read counts) between homozygous mutant mice (on the x-axis) and wt controls (y-axis). Data accompanying all plots are shown in Table S4. **(B)** Comparison between $\log_2$ fold change in circRNA expression and the corresponding fold change of the linear transcript expression from the corresponding parental gene in homozygous mutant mice and wt controls. DE, differential expression; FC, fold change; LFC, log fold change; R1, *Rims1*; R2, *Rims2*. Assorted circRNAs are highlighted by diamond symbols, and their specifications are provided in Table S5. Triangles represent circRNAs with values reaching beyond the displayed scale. Only circRNAs with mean-normalized BSJ read counts minimum of 3 are shown in all plots. **(C)** Statistical model–based estimates of the relative expression of alternative exons, and circRNAs and their corresponding linear transcripts in cerebellar samples collected from 12-wk-old animals from the $Prpf8^{Y2334N/Y2334N}$ strain and 8-wk-old $Prpf8^{\Delta17/\Delta17}$ mice. The estimates are based on a linear mixed model of the RT–qPCR data. Posterior distribution of the ratio of each product (circRNA or *Ntm* alternative exons; horizontal axis) to a corresponding canonical product (linear mRNA

mmu_circ_0011821 (Fig 5A and B), because their parental genes are physiologically enriched in granule and Golgi cells. The linear-to-circular ratio was distorted in favor of the circRNA form (Fig 5A and B); however, parallel quantification by RT–qPCR did not suggest that the circularization was carried out at the expense of linear splicing (Fig S12). Interestingly, in advanced stages of cerebellar neurodegeneration, the expression of Rbfox1, Zfp521, and Chd7 host mRNAs was substantially decreased (Fig S12), in contrast to the surplus of the circRNA forms (Fig 4C).

To further test a role of Prpf8 mutation in circRNA expression, we established three reporters derived from mouse gene loci producing circRNAs (Fig 5C) and expressed them in human RPE cell line containing the Y2334N mutation in both *PRPF8* alleles (N/N). Comparing the expression of circular with linear RNA form, we detected a significant reduction in circRNA generated from the Kmt2a reporter in a RPE$^{N/N}$ cell line (Fig 5D). Consistently, circRNA expressed from the *Kmt2a* (ENSMUSG00000002028) gene was down-regulated in mutated animals (Table S4). These data further show that mutations in the Prpf8 protein affect the expression of particular circRNAs.

To investigate a potential contribution of misregulated splicing of protein-coding genes, we analyzed splicing of linear mRNAs in RNA-Seq runs from 4-wk-old animals. Deregulated splicing involved intron retention, exon skipping, and alternative 5'/3'-splice site recognition. Only events with ≥15% difference between the mutant mice and wt controls were considered biologically relevant (Fig S13 and Table S8). In the *Prpf8$^{Δ17/Δ17}$* animals, we observed large numbers of intron retention events and 25 of those genes overlapped between *Prpf8$^{Δ17/Δ17}$* and *Prpf8$^{Y2334N/Y2334N}$* animals. A more detailed analysis revealed that the same intron was targeted in only one gene, CDH18, but the effect of Prpf8 mutation was opposite, meaning that the targeted intron was better spliced in *Prpf8$^{Δ17/Δ17}$* animals but less efficiently spliced in *Prpf8$^{Y2334N/Y2334N}$* mice. A comparison of exon skipping with alternative 5'/3'-splice site usage did not reveal any overlap between the two mouse strains (Table S8). We further subjected the differentially spliced genes to gene set enrichment analysis. However, we did not identify any overlapping categories between the two mutant strains (Table S9).

Altogether, animals carrying aberrant variants of splicing factor Prpf8 displayed distorted splicing and expression of several circRNAs that are presumably granule cell–specific, and these RNA-processing defects preceded the onset of pathological changes affecting the cerebellar tissue.

## Cerebellar aging is associated with down-regulation of splicing proteins including Prpf8

The expression of spliceosomal components has been previously shown to vary in different developmental stages and on the course to full adulthood (Cao et al, 2011). Here, we took advantage of our mouse models and test the hypothesis that the lower expression of Prpf8 correlates with the onset of the neurodegeneration. To examine the abundance of Prpf8 during and after cerebellar postnatal maturation, we monitored the levels of Prpf8 and three other splicing factors Prpf6, Prpf31, and Snrnp200 in the cerebellum, retina, and control liver snips (Fig 6). We harvested samples from the organs in postnatal weeks 1, 2, 4, and 8, and observed a significant decline in all tested splicing components in the cerebellum from week 4 onward. A similar reduction in splicing protein expression was observed in the retina, but here, the expression of Prpf8 declined to 41%, whereas in the cerebellum, Prpf8 protein levels dropped more than five times to 14% of the amount observed in the first week. Down-regulation of three of four tested spliceosomal components (Prpf8, Prpf6, and Prpf31) was highest in aging cerebellum followed by retina samples. In livers, only Prpf31 protein declined below 50% at week 8. The down-regulation of splicing factors was even more pronounced in animals expressing the RP variants, which indicates that RP mutations, namely, the Y2334N substitution, negatively affect stability of splicing proteins and might contribute to the neurodegeneration phenotype (Figs 7 and S15A).

In parallel to the measurement of Prpf8 protein levels, we also investigated the expression of *Prpf8* RNA in 4-wk-old mice of both strains, and in 8- and 12-wk-old *Prpf8$^{Δ17}$* and *Prpf8$^{Y2334N}$* animals, respectively. The *Prpf8* transcripts were quantified in the RNA-Seq datasets and then verified in independent samples by the RT–qPCR approach. The results surprisingly revealed similar or slightly enhanced abundance of *Prpf8* mRNA in *Prpf8$^{Y2334N/Y2334N}$* and *Prpf8$^{Δ17/Δ17}$* animals in all the examined timepoints when compared to levels of wt *Prpf8* (Fig S14). These findings indicated that the amounts of Prpf8 in cerebellar cells are primarily regulated at the protein level.

To further analyze the relationship between RNA and the protein expression of Prpf8, we took advantage of a mouse strain *Prpf8$^{Δ366}$* that carries a large deletion of 366 bp that eliminated the 3' part of *Prpf8* exon 37 until introns 38 and 39 (chr11:75,506,472–chr11:75,506,837; Fig 1). Because no *Prpf8$^{Δ366/Δ366}$* individuals were born to heterozygotic breeding pairs, the *Prpf8$^{Δ366}$* allele is genetically a null variant, and this finding was in agreement with embryonic lethality previously described for full *Prpf8* deficiency (Graziotto et al, 2011). We did not detect any productive expression originating from the *Prpf8$^{Δ366}$* allele, indicating a rapid decay of the faulty mRNA. In cerebella harvested from *Prpf8$^{Δ366/wt}$* rodents, the *Prpf8* mRNA content reached 60% of the control animals, which strongly argued against a substantial compensatory transcription from the *Prpf8* wt allele (Fig 8A). This shortage in *Prpf8* transcription was, however, compensated at the level of Prpf8 protein synthesis and/or stability, because no differences were recorded in Prpf8 protein abundance estimated by Western blotting and immunohistochemistry (Figs 8B–D and S15B). Consistently with the unchanged expression of the Prpf8 protein, the absence of one *Prpf8* copy did not provoke any pathological changes affecting the cerebellum (Fig 8D), and the *Prpf8$^{Δ366/wt}$* animals attained body weight equivalent to their wt littermates (Fig S16A). These results demonstrated that loss of one *Prpf8* allele per se did not induce cerebellar degeneration

transcript) was computed for each genotype. Estimates of the ratio of those ratios across genotypes are shown (vertical axis, log scale). Values below 1 indicate a lower proportion of the circRNAs or decreased inclusion of the alternative exons, respectively, in the mutant animals compared with wt counterparts. Shown are posterior credible intervals (lines: 95%—thin; 50%—thick), and means (points). Asterisks indicate comparisons for which the 95% credible interval excludes 1 (no difference). The comparison "Rims2 circ_0595 alt. exon A'*" measures the inclusion of alternative exon A' into the *Rims2* circ_0000595.

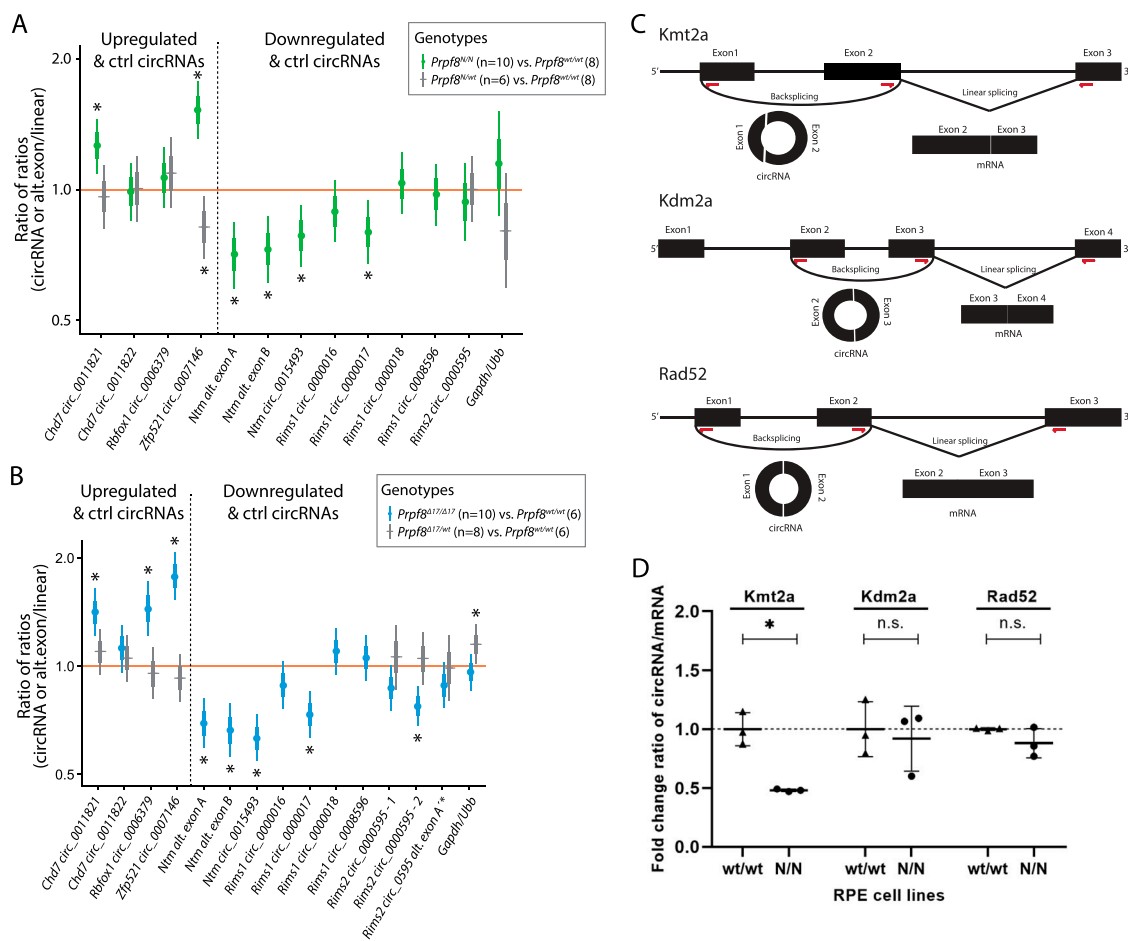

**Figure 5. Differential circRNA expression in 4-wk-old cerebella of both Prpf8 mutant strains and cells expressing PRPF8^Y2334N.**
**(A, B)** Statistical model–based estimates (see Fig 4 for details) of the relative expression of circRNAs and their corresponding linear transcripts in cerebellar samples gained from 4-wk-old animals from the *Prpf8^Y2334N* strain (A), or 4-wk-old *Prpf8^Δ17* mice (B). Shown are posterior credible intervals (lines: 95%—thin; 50%—thick) and means (points). Asterisks indicate comparisons for which the 95% credible interval excludes 1 (no difference). **(C)** Schematic representation of circRNA-producing reporters. Red arrows indicate positions of primers used for detection of circular and linear forms. **(D)** circRNA reporters were transfected into human RPE expressing GFP-PRPF8^WT (wt/wt) or GFP-PRPF8^Y2334N (N/N) from both alleles, and circRNA:linear RNA ratio was determined by RT–qPCR. N = 3. Statistical significance was determined by a paired *t* test. The expression of mRNAs and circRNA based on RNA-Seq data is presented in Fig S11A and B and Tables S2 and S4.

## Discussion

Spliceosomopathies are inborn disorders, where congenital mutations impair generic factors involved in spliceosome assembly and ruled out haploinsufficiency as a mode of action in the case of murine aberrant Prpf8. However, the physiological drop in Prpf8 abundance that coincides with cerebellar postnatal maturation, with ongoing synaptogenesis, and with the onset of circRNA formation suggests that reduced levels of Prpf8 and other splicing factors may sensitize granule cells to Prpf8 mutations. Interestingly, analysis of the *Prpf8^Δ366* strain breeding records revealed a potential distortion of paternal allele transmission (Fig S16B). This observation may indicate that deficiency in Prpf8 and hence execution of splicing can impact the haploid stages of spermatogenesis, or alternatively, the Prpf8 protein can execute unknown extraspliceosomal functions as suggested for PRPF31 (Buskin et al, 2018).

and/or function, but the primary phenotypes manifest in distinct organs. Such selective organ impairment indicates that development and/or homeostasis of certain tissues is specifically sensitive to mutations that are elsewhere tolerated. In addition, the dissimilar outset dynamics imply that particular cell types are likely vulnerable only during a defined developmental or age window. This suggests that naturally occurring changes in the intracellular environment can create conditions for an outbreak of a pathology, but mechanisms that trigger pathological changes are unknown. In this study, we established and analyzed two aberrant variants of splicing factor Prpf8 in a mouse experimental model and observed that both strains displayed rapid and specific decay of cerebellar granule cells. Surprisingly, the other cerebellar cells (e.g., Purkinje cells) were not affected and were able to survive despite significant granule cell decay, at least during the course of the experiment. Survival of other cell types in the cerebellum thus suggests that the effect is cell-autonomous and reflects defects in intracellular homeostasis of granule cells. It raises questions why mouse

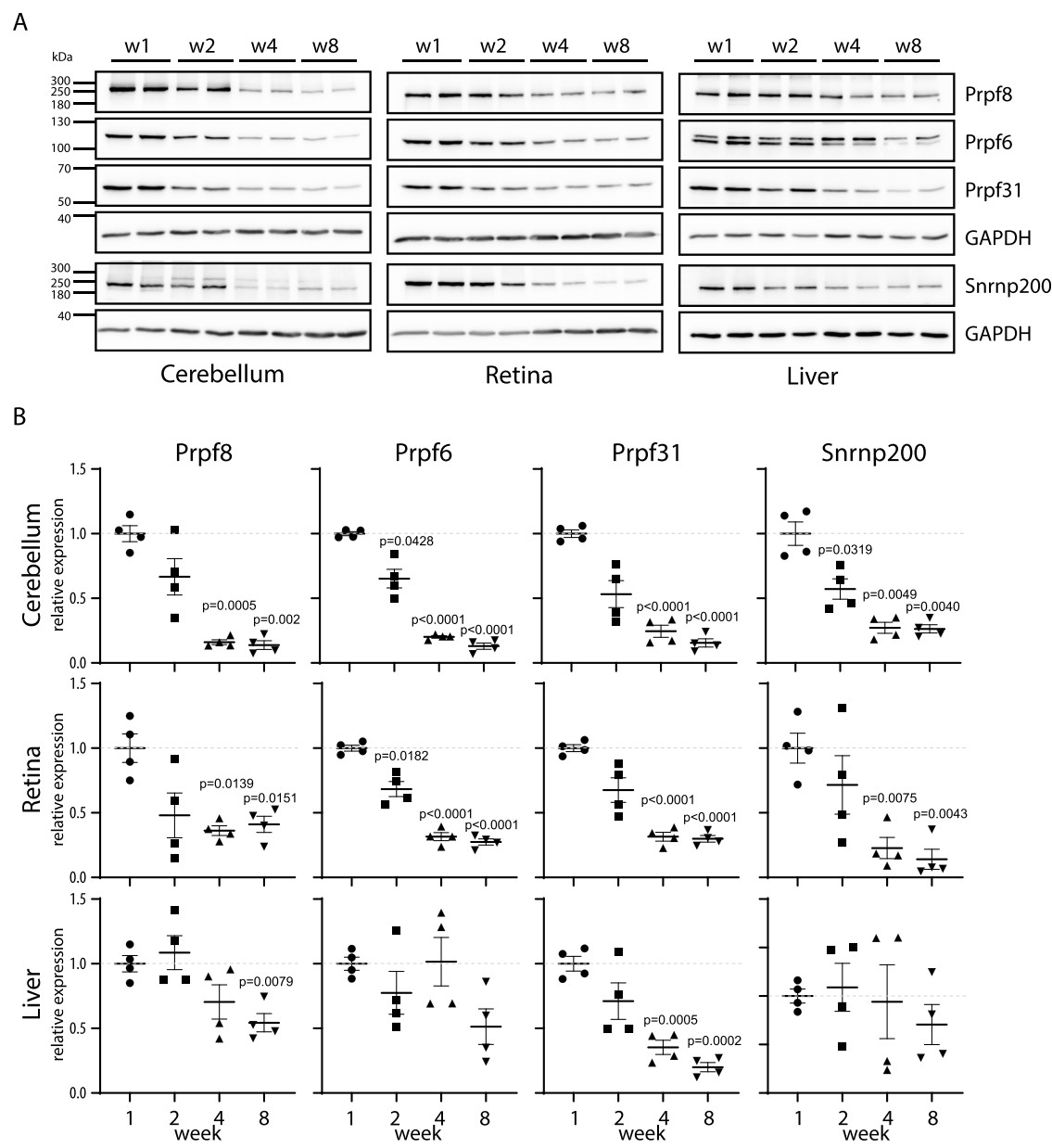

**Figure 6. Expression of snRNP proteins is reduced in mature cerebellum.**
**(A)** Expression of snRNP-specific proteins was assayed in cerebella, retina, and liver of WT animals at 1, 2, 4, and 8 wk by Western blotting. Samples from two animals were analyzed per each timepoint. **(B)** Statistical analysis of protein expression at 1, 2, 4, and 8 wk. Protein abundance in individual animals was first normalized to levels of GAPDH in the same tissue and then calculated as % of average value of the animals at the first week. N = 4. Statistical significance of differential protein abundance in the given genotype groups was examined by one-way ANOVA followed by Dunnett's T3 test to find samples that significantly differ from expression in week 1 (mean with SEM is displayed together with *P*-values).

granule cells are particularly vulnerable to the mutation in a protein essential for all cells? And why we had to breed mice to homozygosity when in humans the disease is autonomously dominant? We have only partial answers and hypothesis to answer these critical questions. It seems that mice can better tolerate mutations in splicing factors. Similar to our findings, mice that genocopy RP substitutions *Prpf3Thr494Met* and *Prpf8His2309Pro* had to be also challenged to homozygosity to initiate significant pathological changes in the RPE layer (Graziotto et al, 2011). In contrast, an

aberrant phenotype in RPE was observed in heterozygous *Prpf31Ala216Pro/wt* rodents (Valdes-Sanchez et al, 2019). However, the p.Ala216Pro substitution in human PRPF31 weakens interactions with U4/U6 snRNP and enhances interaction with U5 snRNP-specific PRPF6 (Huranova et al, 2009). This might inhibit the assembly of the functional U4/U6·U5 tri-snRNP and the mutation might thus have a dominant negative effect.

To better understand the dynamics of splicing factors expression in the targeted tissue, we analyzed the amount of selected key

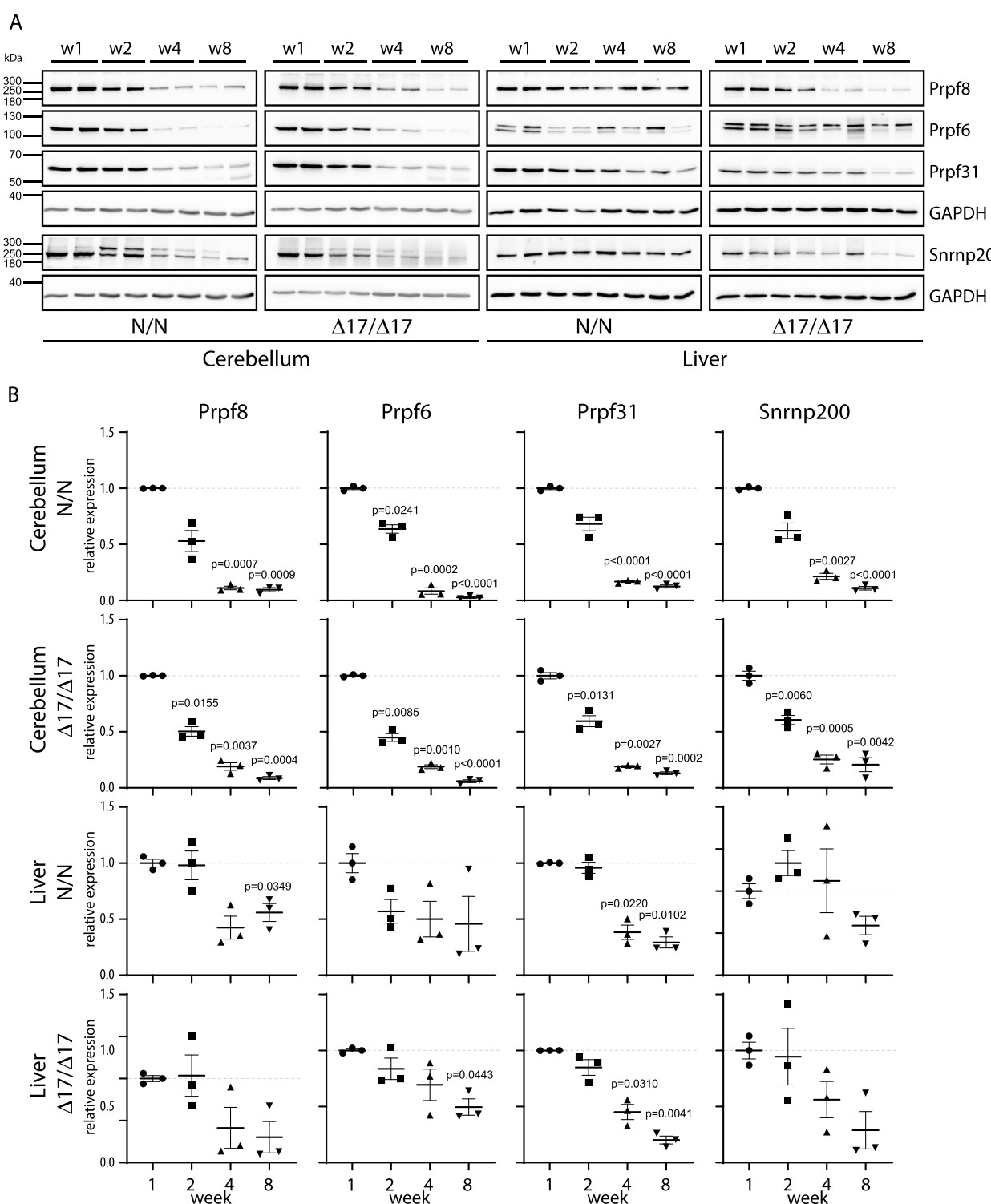

**Figure 7. Expression of snRNP proteins is reduced in mutant animals.**
**(A)** Expression of snRNP-specific proteins was assayed in cerebella and liver of *Prpf8*[Y2334N/Y2334N] and *Prpf8*[Δ17/Δ17] mice at 1, 2, 4, and 8 wk by Western blotting. Samples from two animals were analyzed per each timepoint. **(B)** Statistical analysis of protein expression at 1, 2, 4, and 8 wk. Protein abundance in individual animals was first normalized to levels of GAPDH in the same tissue and then calculated as % of average value of the animals at the first week. N = 3. Statistical significance of differential protein abundance in the given genotype groups was examined by one-way ANOVA followed by Dunnett's T3 test to find samples that significantly differ from expression in week 1 (mean with SEM is displayed together with *P*-values).

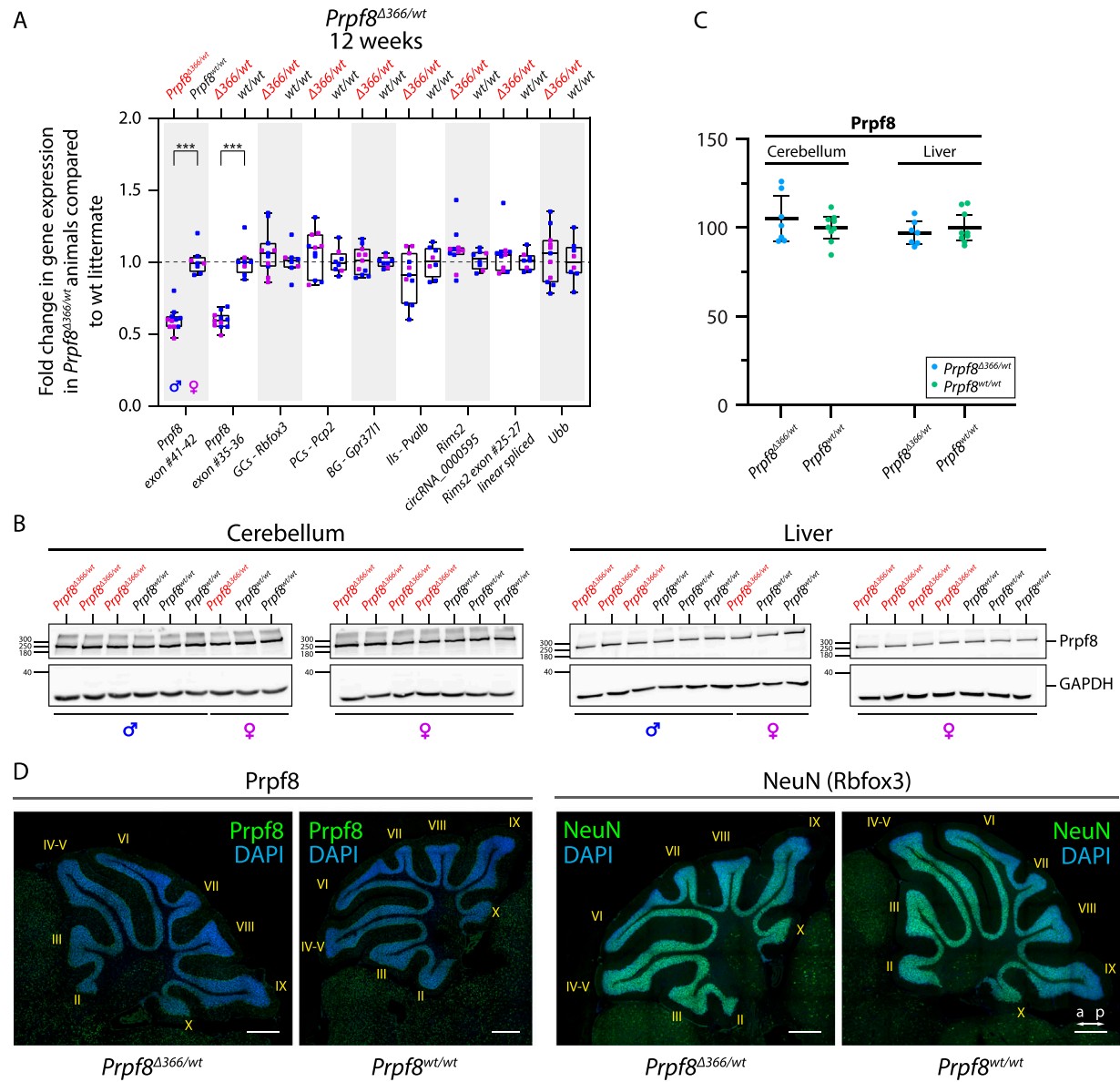

**Figure 8. 12-wk-old *Prpf8^Δ366/wt* mice do not exhibit any degenerative phenotype.**
**(A)** RT–qPCR-based quantification of selected transcripts in the cerebellum of 12-wk-old *Prpf8^Δ366/wt* mice and their *Prpf8^wt/wt* littermates. GCs, granule cells; PCs, Purkinje cells; BG, Bergmann glia; IIs, inhibitory interneurons. **(B)** Western blots from cerebellar and liver samples collected from 12-wk-old *Prpf8^Δ366/wt* mice and their wt littermates. **(C)** Densitometric quantification of Western blots from two *Prpf8^Δ366* cohorts (*Prpf8^Δ366/wt* n = 7 and *Prpf8^wt/wt* n = 9) did not reveal any significant differences in Prpf8 protein levels in *Prpf8^Δ366/wt* animals in neither cerebellar nor liver specimen. Protein abundance of Prpf8 in individual animals was first normalized to levels of GAPDH in the same tissue and then calculated as % of average value of wt animals. Statistical significance of differential protein abundance in the given genotype groups was examined by a *t* test using GraphPad software (displayed is the mean with 95% credible interval). **(D)** Immunohistochemical analysis of Prpf8 and NeuN abundance in cerebella of 12-wk-old *Prpf8^Δ366/wt* animals and their corresponding *Prpf8^wt/wt* counterparts. Representative images are shown for each genetic condition. Nuclear counterstaining by DAPI; scale bar = 200 μm. a, anterior; p, posterior. For further characterization of *Prpf8^Δ366/wt* animals, see Fig S16.

splicing proteins including Prpf8 during the first 8 wk of life in three different tissues of WT animals. We observed a steep and significant drop in all tested splicing proteins in the cerebellum and retina between week 1 and week 4. Moreover, most of the tested splicing proteins were more down-regulated in the cerebellum than in the retina. This physiological reduction in splicing components precedes the onset of neurodegeneration, and we speculate that this could be one of the main reasons why the mouse cerebellum is

specifically sensitive to mutations in Prpf8. In mice, we found higher levels of splicing proteins in the retina than in the cerebellum, which could explain why granule cells, which represent the majority of cerebellar cells, are the primary target in the mouse model. The amount of the PRPF8 protein in the human retina is not available in the Human Protein Atlas (www.proteinatlas.org), and we thus cannot directly compare protein expression in the human retina and cerebellum to evaluate the hypothesis that the lower

expression of splicing factors is associated with cell sensitivity to retinitis pigmentosa mutations.

However, this hypothesis is consistent with the fact that most RP-linked mutations in splicing components reduce their ability to form functional spliceosomal snRNPs, which further decreases the number of splicing-competent complexes inside the nucleus (Gonzalez-Santos et al, 2008; Huranova et al, 2009; Tanackovic et al, 2011b; Linder et al, 2014; Malinova et al, 2017). However, neither of the two Prpf8 aberrant mutations studied in this work affect the Prpf8 interaction with Prpf6 and Snrnp200 and snRNP maturations (Malinova et al, 2017, and Fig S1C). It suggests that the C-terminal tail of PRPF8, which is inserted into and blocks the RNA tunnel in SNRNP200, does not considerably contribute to the interaction between PRPF8 and SNRNP200 and overall stability of the U5 snRNP (Mozaffari-Jovin et al, 2013). It should be noted that two pathological missense mutations in the RNA channel of SNRNP200 did not either inhibit the assembly of snRNPs, but rather had a negative effect on the selection of correct splice sites (Cvackova et al, 2014). The two previously studied mutations in SNRNP200 (p.Ser1087Leu and p.Arg1090Leu) and the substitution p.Tyr2334Asn in PRPF8 all cluster inside the RNA entry channel of SNRNP200 (Mozaffari-Jovin et al, 2013). It is thus plausible that all three RP mutations negatively affect the tight regulation of SNRNP200 helicase activity and thus change splicing of sensitive genes including back-splicing and production of circRNAs. Most cells are able to tolerate this partial malfunction because of the natural high expression of splicing proteins. However, cells and tissues with the physiologically low expression of these proteins are specifically susceptible to mutations in the splicing machinery.

Our data indicate that the regulation of expression of splicing proteins in the cerebellum occurs at the protein level because we observed that the dosage compensation for the missing *Prpf8* allele occurs at the protein but not at the mRNA level (Fig 8). Therefore, monitoring mRNA levels is not the best indicator of actual splicing protein expression, at least in some tissues. A similar observation was done in *Drosophila*, where the overexpression of transgenic Prpf8 did not markedly increase the overall protein abundance (Stankovic et al, 2020). In contrast, a partial reduction in Prpf3 protein levels was observed in Prpf3$^{+/-}$ heterozygotic mice (Graziotto et al, 2008), leaving the regulation of splicing protein expression in neuronal tissues open.

Next, we analyzed how RP mutations alter transcriptome in the targeted tissue to identify common defects between the two established RP-mimicking strains. We did not identify any significant overlap in terms of intron retention, exon skipping, and alternative 5′- and 3′-splice site usage in 4-wk-old cerebella of *Prpf8*$^{Δ17/Δ17}$ and *Prpf8*$^{Y2334N/Y2334N}$ mice. However, it should be noted that only events with a higher than 15% shift were considered and we cannot exclude that minor changes in the splicing pattern of particular mRNAs are common to both Prpf8 RP variants. In contrast, we observed perturbation of specific subclasses of circRNAs in both strains that preceded the onset of neuronal death. Because circRNAs are formed by the splicing machinery, deregulation of circRNA might be a direct effect of Prpf8 mutations. Consistently, inhibition of spliceosome activity was previously shown to disrupt the circRNA pool (Starke et al, 2015; Wang et al, 2019), and down-regulation of Prpf8 in *Drosophila* increased the expression of two model circRNAs (Liang et al, 2017).

Circular RNAs are highly abundant in neural tissues, and cerebellar granule cells are substantially rich in circRNAs (Rybak-Wolf et al, 2015), yet their biological roles remain poorly characterized. Some circRNAs were proposed to regulate gene expression by interfering with miRNA and splicing pathways (Hansen et al, 2013; Ashwal-Fluss et al, 2014; Yu et al, 2017). Transcription of numerous circRNAs, including *Rims2* circ_0000595, was found to be uncoupled from the parental mRNAs (Rybak-Wolf et al, 2015, and this work), which suggests that the circRNA production is regulated independently of the spatial and temporal dynamics of the host gene expression. Indeed, the formation of the *Rims2* circ_0000595 was in vitro induced by neuronal maturation, and in the developing mouse brain, the circ_0000595 amounts substantially increased around P10 to peak in the adulthood (Rybak-Wolf et al, 2015). This time window coincides with establishment of connections between mossy fibers and granule dendrites, and between parallel fibers and Purkinje cells (Kano & Watanabe, 2019). In agreement, many circRNAs were shown to change their abundance in response to the onset of synaptogenesis (You et al, 2015), suggesting the circRNA species likely contribute to establishment and/or maintenance of synapses. Accordingly, circRNAs and specifically the *Rims2* circ_0000595 were found enriched in the brain synaptosome (Rybak-Wolf et al, 2015). Genetic ablation of the cerebellar most prominent circRNA *Cdr1os* mmu_circ_0001878 provoked excitatory synapse malfunction (Piwecka et al, 2017; Kleaveland et al, 2018), and specific elimination of the *Rims2* circRNA by a shRNA approach evoked retinal neuron apoptosis (Sun et al, 2021). Together, circRNAs likely represent components of a complex network neurons use to fine-tune their transcriptome and to modulate synaptic events, and their disturbance might compromise neuronal fitness.

What can the newly established mouse models tell us about retinal degeneration in humans? In mice, the Rims proteins are localized not only to presynaptic active zones in conventional synapses, but also to presynaptic ribbons in photoreceptor ribbon synapses (Wang et al, 1997). Consistently, inborn errors in human *RIMS1* and *RIMS2* underlie cone–rod dystrophies (OMIM #603649 and #618970) (Johnson et al, 2003; Mechaussier et al, 2020). However, it is not known whether the *RIMS* mutations may simultaneously impact the biogenesis of cognate circRNAs. Retina belongs to neuronal tissues expressing a high amount of various circRNA species; many of them are conserved among humans and mice including *RIMS1* hsa_circ_0132250 and RIMS2 hsa_circ0005114 (corresponding to the murine *Rims 1* and *Rims2* circRNAs, respectively [Fig S16C and D]), and circHIPK3 (Izuogu et al, 2018; Mellough et al, 2019; Sun et al, 2019; Rahimi et al, 2021). Down-regulation of the circHIPK3 accelerated apoptosis in cultured lens epithelial cells (Liu et al, 2018), which indicates that circRNAs indeed play an important role in eye homeostasis. Overall, the novel mouse models of aberrant Prpf8 suggest that deregulation of circRNAs could represent a pathological mechanism that co-provokes the retinal dystrophy in splicing factor RP.

In summary, we suggest that vulnerability of the murine cerebellum toward mutant Prpf8 proteins probably originates from a physiological drop in splicing factor abundance that coincides with ongoing processes of tissue maturation and circRNA-mediated synaptogenesis. Deregulation of circular RNAs may represent a

pathological mechanism that perturbs cellular homeostasis and initiates apoptosis of vulnerable cells. Finally, the infiltration of microglia and astrocytes can further intensify and complete the damage of the neuron tissues.

## Materials and Methods

### Generation, breeding, and genotyping of mutant mice

All animal models and experiments of this study were ethically reviewed and approved by the Animal Care Committee of the Institute of Molecular Genetics (Ref. No. 45/2020). All novel alleles of *Prpf8* were established in the Transgenic and Archiving Module of the Czech Centre for Phenogenomics (BIOCEV). The pathogenic substitution Prpf8 Tyr2334Asn was introduced to the exon 42 of the *Prpf8* gene using Cas9-mediated HDR; the short guide RNA (sgRNA) specifically targeting the vicinity of Tyr2334 (*Prpf8* ex42 sgRNA-#1) was devised with assistance of the MIT Design website (https://crispr.mit.edu) (Ran et al, 2013). The *Prpf8* ex42 sgRNA-#1 was assembled and its efficacy first assayed in vitro. In detail, complementary guide oligodeoxynucleotides were first phosphorylated by T4 Polynucleotide Kinase (Thermo Fisher Scientific) in ATP-containing T4 DNA Ligase buffer (Thermo Fisher Scientific), then denatured for 5 min at 95°C, and annealed by gradual cooling down at −0.1°C/s using a PCR thermocycler (T100; Bio-Rad). Annealed oligos were subsequently inserted into the BbsI-digested vector pX330-U6-Chimeric_BB-CBh-hSpCas9 (#42230; Addgene), where the co-expression of sgRNA and human codon-optimized *S.pyogenes* Cas9 is driven from human U6 promoter and CAGEN promoter/enhancer, respectively (Cong et al, 2013). Efficacy of the *Prpf8* ex42-#1 sgRNA toward the target sequence in *Prpf8* (5′-AAGGAAGT AACTAGGCATAG-AGG; 11:75,509,277–75,509,299 coding strand) and two predicted, highest scoring off-target loci (chr.11 5′-AAGGAACTA ACTAGGCATGG-AGG 11:5,006,551–5,006,573; and chr.14 5′-AAGAAAGA AACTTGGCATAG-CAG 14:56,271,668–56,271,690) was assessed in transfected HeLa cells using a TurboRFP open reading frame reconstitution reporter plasmid (pAR-TurboRFP; #60021; Addgene) as described previously (Kasparek et al, 2014).

To prepare individual components for the microinjection mixture, Cas9 was transcribed from a linearized parental pX330 plasmid using the mMESSAGE mMACHINE T7 kit (Ambion/Thermo Fisher Scientific); the resulting mRNA was polyadenylated using the Poly(A) Tailing kit (Ambion/Thermo Fisher Scientific) and purified with the RNA Clean and Concentrator kit (Zymo Research). PCR template for in vitro sgRNA synthesis included T7 promoter, 20-nt sgRNA, and tracrRNA sequences and was amplified from a pX330-*Prpf8* ex42-#1 sgRNA plasmid using T7 promoter–containing and tracrRNA primers. *Prpf8* ex42-#1 sgRNA was subsequently produced by MEGAshortscript T7 Transcription Kit (Ambion/Thermo Fisher Scientific) and purified using the ssDNA/RNA Clean and Concentrator kit (Zymo Research). A synthetic oligodeoxynucleotide or double-stranded DNA fragment encoding the mutated Tyr2334Asn motif was used in parallel as potential HDR donors to navigate the HDR machinery. Briefly, the donor oligodeoxynucleotide contained 35-bp homology arms flanking a central region where the sequence

encoding for five Prpf8 C-terminal residues spanning Glu2331 to STOP codon was silently mutagenized to prevent re-editing of once-modified allele (Krchnakova et al, 2019). These substitutions were deliberately designed to introduce novel DraI and MluI restriction sites that allowed downstream genotypic screening by RFLP. In addition, the donor oligodeoxynucleotide contained three terminal linkages at both 5′ and 3′ that are modified to phosphorothioate. The HDR oligo was purchased from Sigma-Aldrich (vendor purification by HPLC) and was also purified using the ssDNA/RNA Clean and Concentrator kit (Zymo Research). In contrast, the double-stranded DNA HDR donor was prepared in a step-wise manner in the pGEM-T Easy vector (Promega) by first cloning in ~800-bp-long homology arms that were amplified from C57BL/6N-derived genomic DNA (left arm: 799 bps upstream of Tyr2334; right arm: 864 bps downstream of Tyr2334; total insert length 1,666 bps, chr11: 75,508,482–75,510,147). We subsequently used mutagenic PCR to alter the vicinity of Tyr2334 with silent substitutions that were identical to those present in the HDR oligo. To prepare the final dsDNA fragment destined for the microinjection, the donor portion was amplified from the pGEM-T Easy Prpf8 Y2334N vector using Phusion High-Fidelity DNA Polymerase (Thermo Fisher Scientific); the sample was then treated with DpnI to remove the parental plasmid and purified using the DNA Clean & Concentrator kit (Zymo Research). Before microinjection, the exact sequences of both the oligo and dsDNA fragments were verified by Sanger sequencing. The microinjection sample comprised Cas9 mRNA (100 ng/µl final concentration), sgRNA (50 ng/µl), and either dsDNA fragment (5 ng/µl) or 10 µM oligodeoxynucleotide. The final mixture was filter-sterilized by passing through a 0.22-µm Millex-GV PVDF filter (Millipore) and microinjected into pronuclei of C57BL/6N-derived zygotes, and those were transferred into pseudo-pregnant recipient mice.

The *Prpf8 Δ366* strain was obtained secondary to genome editing effort aimed at residue Ser2118 in exon 38 of the *Prpf8* gene. TALENs were designed using TAL Effector Nucleotide Targeter 2.0 (talent.cac.cornell.edu [Doyle et al, 2012]), assembled using the Golden Gate Cloning protocol, and inserted into the ELD-KKR backbone plasmids as described previously (Kasparek et al, 2014). The DNA binding domains of TALEN comprised the following repeats: NN-HD-NG-NG-NI-NI-NN-NI-NI-NN-NG-NG-HD-NI-NG-HD-NG (5′-TALEN-Prpf8 ex38) and NI-NN-HD-NG-NI-HD-NG-HD-NI-HD-NG-NG-NN-NN-NN-HD-NI-HD (3′-TALEN-Prpf8 ex38). Linearized plasmids were in vitro–transcribed using the mMESSAGE mMACHINE T7 kit (Ambion/Thermo Fisher Scientific); the resulting mRNA was poly-adenylated using the Poly(A) Tailing kit (Ambion/Thermo Fisher Scientific) and purified with the RNA Clean and Concentrator kit (Zymo Research). TALEN-encoding mRNA (20 ng/µl each) was mixed with targeting single-stranded oligodeoxynucleotide (10 µM final concentration; Sigma-Aldrich), and the final solution was filtered through a 0.22-µm Millex-GV PVDF filter before microinjection into C57BL/6N-derived zygotes.

In the IMG mouse, husbandry animals were maintained under a 12/12-h light cycle, and access to food and water was provided ad libitum. All novel *Prpf8* strains were generated in zygotes isolated from the C57BL/6N substrain and successively back-crossed to the C57BL/6J background to eliminate the *Crb1^rd8* mutation that may else confound interpretation of ocular phenotypes (Mattapallil

et al, 2012). Phenotypic analysis was carried out after seven generations of breeding to C57BL/6J, when contribution of the recurrent parent genome is >99% (Visscher, 1999). Routine PCR genotyping was performed with tail biopsies snipped from 3-wk-old animals on weaning. Tissue sample was incubated overnight at 56°C in 100 $\mu$l of lysis solution (10 mM Tris-Cl, pH 8.3, 50 mM KCl, 0.45% [vol/vol] Nonidet P-40 Substitute, 0.45% [vol/vol] Tween-20, and 0.1 mg/ml gelatin from porcine skin; Sigma-Aldrich) supplemented with Proteinase K (0.2 mg/ml; Thermo Fisher Scientific), then heat-inactivated for 15 min at 70°C, and 1 $\mu$l of the crude lysate was directly used as input for PCRs (DreamTaq Green PCR Master Mix [Thermo Fisher Scientific] supplemented with 1 M betaine and 3% DMSO [Sigma-Aldrich]). To increase the specificity of the genotyping primers, an extra mismatch was deliberately introduced at the third position from the 3′ end of one of the primers; primers were designed using the PRIMER1 ARMS-PCR tool (primer1.soton.ac.uk/primer1.html). In contrast, RFLP was performed with amplicons produced by DreamTaq DNA polymerase (Thermo Fisher Scientific), which were purified by sodium acetate–ethanol precipitation, and then subjected to restriction digestion with a MluI enzyme (New England Biolabs) to identify mutant animals. To reveal the spectrum of Prpf8 exon 42 genetic variants present in the founder animals, RFLP amplicons were cloned into the pGEM-T Easy vector and subsequently analyzed by Sanger sequencing. The list of all primers relevant to mouse generation and breeding is shown in Table S1.

## Preparation of RPE edited cell lines

To introduce the Y2334N mutation and add the GFP tag (or add the GFP tag only), we first designed a guide RNA sequence targeting the very C-terminus of PRPF8 to be edited using the online CRISPR design tool (Hsu et al, 2013). The guide RNA was cloned into the pX330-U6-Chimeric BB-CBh-hSpCas9 plasmid (# 42230; Addgene) using BbsI restriction sites. To test the efficiency of the guide RNA, we cloned the guide target sequence into the pARv-RFP reporter plasmid (plasmid # 60021; Addgene) using EcoRV and PvuI restriction sites with the introduction of the BamHI restriction site at 5′ end of a guide target sequence to allow restriction digest analysis of positive clones. The HDR template was assembled from left and right homology arms (of 1,500 bp each) that flank the target sequence (eventually carrying the Y2334N mutation) and GFP sequence with SV40pA and were cloned one by one into the pBluescript II vector. Left and right homology arms were amplified by PCR from genomic DNA, and the target sequences with GFP were amplified from PRPF8-pEGFP-N1 and Y2334N-pEGFP-N1 plasmids. All cloning products were confirmed by DNA sequencing. HDR template was amplified by PCR from the pBluescript II vector using T3 and T7 primers, and DNA was purified by precipitation with sodium acetate and ethanol. RPE cells were grown to 90% confluence and co-transfected with linearized HDR template and pX330 harboring the guide RNA sequence using the Lipofectamine LTX Transfection Reagent (Thermo Fisher Scientific) according to the manufacturer's protocol. To prevent the non-homologous end-joining (NHEJ), the NHEJ inhibitor SCR7 (final concentration 1 $\mu$M) was added to cells 1 d before transfection and was freshly added every next day until FACS sorting. Cells were sorted 72 h after transfection for GFP positivity, one cell at a time into the wells of 96-

well plates containing a mixture of fresh and conditioned medium (1:1). Positive clones were selected by PCR using reverse primer within GFP and forward primer upstream of the left homology arm. Finally, the genomic DNA of the edited clone was sequenced to confirm the proper editing of the target sequence and five top off-target sequences were verified being unaffected (see Fig S17 for characterization of RPE cells carrying either WT PRPF8-GFP (RPE^{wt/wt}) or PRPF8^{Y2334N}-GFP (RPE^{N/N})). Primers used for editing and off-target detection are listed in Table S1.

## Immunohistochemistry

Mice were killed at indicated timepoints by $CO_2$ asphyxiation and *post mortem* perfused with 10 ml PBS followed by 10 ml of 4% PFA (Electron Microscopy Services)/PBS by intracardial puncture. After decapitation, skin removal, and skull opening, the whole head was fixed *en bloc* in 4% PFA/PBS for 24 h and transferred to 70% ethanol (Penta), and then, cerebella were dissected out for subsequent tissue processing (ASP200S; Leica) and paraffin embedding (EG1150H; Leica). 5-$\mu$m sections were stained according to standard protocols. Briefly, specimens were deparaffinized in xylene (Penta) and rehydrated through graded ethanol series; antigen retrieval was performed by boiling in 10 mM sodium citrate buffer, pH 6.0 (Sigma-Aldrich), in pressure cooker for 20 min. Endogenous peroxidase activity was blocked by immersion in 3% $H_2O_2$ (Sigma-Aldrich) in methanol (Penta) for 10 min; interference from endogenous biotin and/or streptavidin binding activity was moreover reduced by the Avidin/Biotin Blocking kit (Thermo Fisher Scientific). Tissue sections were incubated overnight in a humidified chamber with the following primary antibodies: anti-CD45 (ab10558; Abcam), anti-GFAP (#12389; Cell Signaling), anti-IP3R-I/II/III (sc-377518; Santa Cruz), anti-NeuN (#24307; Cell Signaling), anti-Prpf8 (ab79237; Abcam), anti-PSD95 (#3409; Cell Signaling), and anti-S100 beta (ab52642; Abcam). Subsequently, biotin-conjugated secondary antibodies (Biotin-XX Goat anti-Mouse IgG B2763 or Biotin-XX Goat anti-Rabbit IgG B2770; Thermo Fisher Scientific) were used at a dilution of 1:750, and the signal was finally visualized with Alexa Fluor 488–labeled streptavidin (S11223; Thermo Fisher Scientific). Nuclei were counterstained with DAPI (Sigma-Aldrich). Immunofluorescence images were acquired using a Dragonfly Spinning Disc confocal microscope (Andor), deconvolved using Huygens Professional software, and processed using FiJi. Control sections accompanying immunohistochemistry slides were counterstained with hematoxylin solution modified according to Gill II (Penta), followed by 0.5% aqueous solution of Eosin Y (Sigma-Aldrich). Nissl staining was performed with 0.1% acidic cresyl violet solution (Abcam) according to the manufacturer's instructions.

Histopathological analyses were carried out on biopsies harvested from mice euthanized by cervical dislocation. For investigation of eye specimen, skinned heads were first fixed for 48 h in Davidson's fixative before eye removal; other examined tissues were routinely fixed for 24 h in phosphate-buffered 10% formalin and then transferred to 70% ethanol solution. Fixed samples were processed using automated tissue processor (ASP6025; Leica) and embedded in paraffin blocks using a Leica EG1150H embedding station. 2-$\mu$m sections were counterstained with hematoxylin, Harris modified, in combination with Eosin B (Sigma-Aldrich), and

processed in Leica Stainer Integrated Workstation (ST5020-CV5030), in combination with the Leica CV5030 coverslipper. Special stains, including Alcian blue, Congo red, Luxol fast blue, and Masson's trichrome, were performed using the Ventana BenchMark Special Stains platform (Roche). Sections were scanned using a Axio Scan.Z1 microscope (Zeiss) using a 20× Plan-Apochromat objective.

Apoptotic cells were visualized with NeuroTACS II In Situ Apoptosis Detection Kit (Trevigen), and the signal was subsequently enhanced using a combination of Vectastain ABC Kit (Vector Laboratories) and Alexa Fluor 488 Tyramide SuperBoost Streptavidin Kit (Thermo Fisher Scientific).

Composition of animal cohorts collected for staining procedures (all genetic conditions contained animals of both genders) was as follows:

(1) Prpf8 Y2334N strain at 4 wk—histopathology: $Prpf8^{Y2334N/Y2334N}$ n = 1, $Prpf8^{Y2334N/wt}$ n = 4, and $Prpf8^{wt/wt}$ n = 1 (Fig S3A)
(2) Prpf8 d17 strain at 4 wk—histopathology: $Prpf8^{\Delta17/\Delta17}$ n = 3, $Prpf8^{\Delta17/wt}$ n = 4, and $Prpf8^{wt/wt}$ n = 1 (Fig S4A)
(3) Prpf8 Y2334N strain at 6 wk—histopathology: $Prpf8^{Y2334N/Y2334N}$ n = 1, $Prpf8^{Y2334N/wt}$ n = 6, and $Prpf8^{wt/wt}$ n = 1 (Fig S3B)
(4) Prpf8 Y2334N strain at 12 wk—histopathology: $Prpf8^{Y2334N/Y2334N}$ n = 2, $Prpf8^{Y2334N/wt}$ n = 1, and $Prpf8^{wt/wt}$ n = 1 (Fig S3C)
(5) Prpf8 d17 strain at 6 wk—histopathology: $Prpf8^{\Delta17/\Delta17}$ n = 3, $Prpf8^{\Delta17/wt}$ n = 8, and $Prpf8^{wt/wt}$ n = 2 (Fig S4B)
(6) Prpf8 Y2334N strain at 17 wk—histopathology: $Prpf8^{Y2334N/Y2334N}$ n = 4, $Prpf8^{Y2334N/wt}$ n = 6, and $Prpf8^{wt/wt}$ n = 4 (Fig S3D)
(7) Prpf8 d17 strain at 15 wk—histopathology: $Prpf8^{\Delta17/\Delta17}$ n = 4, $Prpf8^{\Delta17/wt}$ n = 5, and $Prpf8^{wt/wt}$ n = 4 (Fig S4C)
(8) Prpf8 Y2334N strain at 22 wk—histopathology: $Prpf8^{Y2334N/Y2334N}$ n = 5, $Prpf8^{Y2334N/wt}$ n = 9, and $Prpf8^{wt/wt}$ n = 5 (Figs S3E and S5A, C, E, and G)
(9) Prpf8 d17 strain at 22 wk—histopathology: $Prpf8^{\Delta17/\Delta17}$ n = 3, $Prpf8^{\Delta17/wt}$ n = 8, and $Prpf8^{wt/wt}$ n = 5 (Figs S4D and S5B, D, F, and H)
(10) Prpf8 d17 strain at 22 wk—histopathology, OCT, ERG: $Prpf8^{\Delta17/\Delta17}$ n = 3, $Prpf8^{\Delta17/wt}$ n = 5, and $Prpf8^{wt/wt}$ n = 7 (Fig S6A–F)
(11) Prpf8 Y2334N strain at 12 wk—Nissl, IHC-P: $Prpf8^{Y2334N/Y2334N}$ n = 6, $Prpf8^{Y2334N/wt}$ n = 7, and $Prpf8^{wt/wt}$ n = 5 (Figs 2B and D and S8)
(12) Prpf8 d17 strain at 8 wk—Nissl, IHC-P: $Prpf8^{\Delta17/\Delta17}$ n = 4, $Prpf8^{\Delta17/wt}$ n = 10, and $Prpf8^{wt/wt}$ n = 2 (Figs 2A and C and S8)
(13) Prpf8 Y2334N strain at 4 wk—IHC-P: $Prpf8^{Y2334N/Y2334N}$ n = 3, $Prpf8^{Y2334N/wt}$ n = 4, and $Prpf8^{wt/wt}$ n = 1 (Fig S10A)
(14) Prpf8 d17 strain at 4 wk—IHC-P: $Prpf8^{\Delta17/\Delta17}$ n = 3, $Prpf8^{\Delta17/wt}$ n = 4, and $Prpf8^{wt/wt}$ n = 1 (Fig S10A)
(15) Prpf8 d17 strain at 5 wk—IHC-P, TUNEL: $Prpf8^{\Delta17/\Delta17}$ n = 4, $Prpf8^{\Delta17/wt}$ n = 6, and $Prpf8^{wt/wt}$ n = 4 (Fig S10B)
(16) Prpf8 Y2334N strain at 5 wk—IHC-P: $Prpf8^{Y2334N/Y2334N}$ n = 2, $Prpf8^{Y2334N/wt}$ n = 3, and $Prpf8^{wt/wt}$ n = 3 (Fig S10C)

## Grip strength measurement

Testing was performed in a room with light intensity set to 110 lux. To assess neuromuscular function, each mouse was gently pulled by tail over metal bars. The grip of only forelimbs and combined forelimbs and hindlimbs was thereby recorded using an automated Grip Strength Meter (Bioseb). The average of three trials was calculated; data are presented as absolute grip strength (g) and normalized to body weight.

## Open field

The open field test evaluates animal motility triggered by exploratory drive in a new environment. Fully automated analysis of animal behavior in open space was based on the video tracking system (Viewer; Biobserve GmbH). The software distinguishes an animal as an object contrasting with the background Video 1. Testing apparatus was uniformly illuminated with the light intensity of 200 lux in the center of the field. Each animal was tested in open field for 20 min. The distance travelled, average speed, and resting time were automatically computed for each 5-min-long interval.

## Electroretinography (ERG)

Animals were dark-adapted overnight in red, individually ventilated cages (Tecniplast). Experiments were carried out in general anesthesia (tiletamine + zolazepam, 30 + 30 mg/kg, i.m.; Virbac). Anesthetized mice were kept on a heating pad, and their eyes were protected against drying using a small amount of transparent eye gel (Vidisic; Bausch & Lomb). Experiments were approved by the Animal Care and Use Committee of the Academy of Sciences of the Czech Republic.

Single-flash stimulation and full-field ERG recording were performed inside a Ganzfeld globe controlled by the RETIanimal system (Roland Consult). Active golden ring electrodes were gently positioned on the cornea, and reference and grounding needle electrodes were placed subdermally in the middle line of the snout and in the back of the animal, respectively. Signal was band-pass-filtered between 1 and 300 Hz and recorded with 1,024 samples/s resolution. Stimulation was repeated several times, and 20–30 individual responses were averaged.

Signals were analyzed using custom scripts in MATLAB (The MathWorks). Amplitudes and latencies (implicit times) of waves a and b were quantified. The oscillatory potential (OP) was extracted from the original recordings by high-pass filtering with a cutoff frequency set to 70 Hz. Amplitude and latency of the four major OP peaks were quantified and then summed. Statistical analysis of data where two parameters were changing, that is, genotype of animals and intensity of the stimulation, was performed by two-way ANOVA, and $P$-values for the factor of genotype were reported.

## OCT

The optical coherent tomograph (OCT, Heidelberg Engineering) scans and quantifies the reflection of a light beam sent from the layers of the retina and composes cross-sectional images of the retina (57 cross-sectional images as minimum). Each cross-sectional image of the retina was evaluated, and the following parameters were measured: the thickness and the gross morphology of the retina, form and position of the optical disc, and the superficial blood vessels and their pattern. The animals were anesthetized using intramuscular injection of tiletamine + zolazepam (30 + 30 mg/kg; Virbac), and the pupils were dilated using eye drops containing 0.5% (wt/vol) atropine (Samohyl).

## Protein extracts and immunoblotting

Dissected cerebella and control liver snips were rinsed in ice-cold PBS, briefly chopped with a razor blade, and then homogenized with a 5-mm generator probe in 0.5 ml of modified RIPA buffer (50 mM Tris-Cl, pH 7.4, 150 mM NaCl, 1% [vol/vol] Triton X-100, 0.5% [wt/vol] sodium deoxycholate, 0.1% [wt/vol] SDS, 1 mM EDTA, 10 mM NaF, 1 mM PMSF, and 1:200 Protease Inhibitor Cocktail Set III [all chemicals purchased from Sigma-Aldrich]). Dissected retinas from both eyes were rinsed with ice-cold PBS and then homogenized in 0.150 ml of modified RIPA buffer. Cell disruption was moreover promoted by sonicating the crude lysate with 30 ultrasonic pulses of 1 s that were interrupted by 1-s pauses using a sonicator (IKA Labortechnik). Protein concentration was determined with Pierce BCA Assay Kit (Thermo Fisher Scientific); loads comprised 20 µg of total protein for liver and retina samples and 30 µg for brain tissue, respectively. Immunoblotting was performed according to established protocols (Malinova et al, 2017), using the following primary antibodies: anti-GAPDH (#5174, 1:1,000; Cell Signaling), anti-NeuN (#24307, 1:1,000; Cell Signaling), anti-Prpf8 (ab79237, 1:1,000; Abcam), anti-Prpf6 (#sc-166889, 1:500; Santa Cruz Biotechnology), anti-Prpf31 (#188577, 1:4,000; Abcam), and anti-Snrnp200 (#HPA029321, 1:250; Sigma-Aldrich). Peroxidase-linked anti-rabbit secondary antibody (#7074, 1:10,000; Cell Signaling) or peroxidase-linked anti-mouse secondary antibody (#115-035-003, 1:10,000; Jackson ImmunoResearch Laboratories) was used in combination with SuperSignal West Femto Maximum Sensitivity Substrate or West Pico PLUS Chemiluminescent Substrate (Thermo Fisher Scientific) to obtain a chemiluminescent signal.

## Immunoprecipitation

Dissected cerebella were rinsed with ice-cold PBS, chopped with a razor blade, and transferred into 1 ml of NET2 buffer (150 mM NaCl, 0.05% NP-40, and 50 mM Tris–HCl, pH 7.4, supplemented with 1:200 Protease Inhibitor Cocktail Set III [#539134; Merck Millipore]), where they were homogenized for 5 s on ice. The crude lysates were then sonicated with 30 consecutive pulses of 1 s (60% amplitude) and cleared by centrifugation at 15,000g for 5 min at 4°C. The cleared lysates were further incubated with 4 µg of Prpf8 antibody (#79237; Abcam) or with 4 µg of control IgG antibody (#I5381; Sigma-Aldrich) for 1 h at 4°C with continuous rotation. 30 µl of Protein G agarose beads (#sc-2002; Santa Cruz Biotechnology) was then added to the mixture, and the reaction was incubated for an additional 2 h at 4°C with continuous rotation. The beads were then washed five times with NET2 buffer and resuspended in 2× sample buffer (250 mM Tris–HCl, pH 6.8, 20% glycerol, 4% SDS, and 0.02% bromophenol blue), supplemented with 24 mM DTT (Sigma-Aldrich). Precipitated proteins were analyzed by Western blotting. Primary antibodies used for the immunoblotting were as follows: anti-GAPDH (#5174T, 1:1,000; Cell Signaling Biotechnology), anti-Prpf6 (#sc-166889, 1:500; Santa Cruz Biotechnology), anti-Prpf8 (#79237, 1:1,000; Abcam), and anti-Snrnp200 (#HPA029321, 1:250; Sigma-Aldrich).

## RT–qPCR

Dissected cerebella (bulk tissue) were briefly rinsed in ice-cold PBS and homogenized in 750 µl of TRIzol reagent (Thermo Fisher Scientific). Total RNA was extracted using Direct-zol RNA Miniprep Kit (Zymo Research) including in-column DNase-I treatment. First-strand cDNA synthesis was performed with Superscript III Reverse transcriptase (Thermo Fisher Scientific) supplemented with random hexamers (Sigma-Aldrich) and RiboLock RNase Inhibitor (Thermo Fisher Scientific). RT–qPCR was carried out in technical triplicates using SYBR Green I Master Mix and LightCycler 480 apparatus (Roche); negative control was represented by cDNA synthesis samples performed in the absence of the reverse transcriptase enzyme. Primers are listed in Table S1.

Statistical evaluation of the RT–qPCR data: for individual genes, the average of threshold cycle (Ct) values from technical triplicates was first normalized to *Gapdh* expression to obtain the ΔCt values. Relative mRNA abundance was then calculated as the fold change (FC) difference of homozygous (hom) or heterozygous (het) animals to $Prpf8^{wt/wt}$ controls using the $2^{\wedge \Delta\Delta Ct}$ approach (i.e., $2^{\wedge \Delta Ct(wt/wt) - \Delta Ct(hom)}$ or $2^{\wedge \Delta Ct(wt/wt) - \Delta Ct(het)}$, respectively). The fold change values are plotted in box-and-whisker diagram according to Tukey with indicated position of median; statistical significance of differential expression in the given genotype groups was examined by a *t* test using GraphPad software (*P < 0.05, **P < 0.01, and ***P < 0.001).

RT–qPCR linear model: to model qPCR data, we used a linear mixed-effects model as outlined in Steibel et al (2009) and Matz et al (2013). Briefly, this method can be understood as an extension of the delta–delta method. The main advantage of mixed-effects models is that they can account for the correlations in the data introduced at multiple levels (replicates from the same animal, samples in the same qPCR runs, and animals from the same litter). In designs involving such correlations, using a mixed-effects model provides better quantification of uncertainty than simpler approaches such as averaging replicates before making comparisons. The models were fitted using the brms R package (Bürkner, 2017). For the experiment with most data points ($Prpf8^{\Delta17}$ at 4 wk), the model included fixed effects of primer, and the full interaction of primer and genotype, and primer and sex. Varying intercepts were then introduced to model variability between qPCR runs, litters, and animals. We also let the residual SD vary between primers and qPCR runs. For other experiments, this full model failed to converge because of the lack of data to inform all the coefficients and some varying intercepts were omitted. We checked that the reduced models did not underestimate the biological and technical variability using posterior predictive checks (Gabry et al, 2019).

To compare the expression of circular/alternative exons with the canonical transcripts, we used the fitted model to compute posterior predictions for the ratio of the circular/alternative exon species to the canonical species within each genotype. We then computed estimates of the ratio of those ratios between the genotypes. In other words, we estimated to what extent does the proportion of the products converted to circular/alternative exon change between the genotypes, which we find more relevant to the current investigation than comparing the absolute expression of the circular/alternative forms. We note that the estimates of this "ratio of ratios" do not depend on the choice of reference, but only on the relative PCR efficiency of the two species. Here, we report results assuming equal efficiency, but almost all of the qualitative patterns are robust to even noticeably different efficiencies (data

not shown). To compare the absolute expression of the linear forms, we used *Gapdh* as reference. Complete code and data to reproduce the qPCR analysis and the detailed outputs of the models are available at https://github.com/cas-bioinf/prpf8

## RNA-Seq

Freshly dissected cerebella (bulk tissue) were briefly rinsed in ice-cold PBS and immediately homogenized in 750 µl of TRIzol reagent (Thermo Fisher Scientific). The homogenizer probe was sanitized between individual specimens to prevent any potential carryover. Total RNA was extracted using Direct-Zol RNA Miniprep Kit (Zymo Research) including in-column DNase-I treatment, and its quality was assessed using the 2100 Bioanalyzer (Agilent); only samples with the RNA integrity number (RIN)>8 were permitted for down-stream processing. Removal of ribosomal RNAs from samples collected from 4- and 8-wk-old $Prpf8^{\Delta 17}$ mice, and 12-wk-old $Prpf8^{Y2334N}$ animals was achieved using RiboCop rRNA Select and Deplete rRNA Depletion Kit v1.2 (Lexogen), and libraries were generated using Total RNA-Seq Library Prep Kit (Lexogen). Cere-bellar specimen gained from 4-wk-old $Prpf8^{Y2334N}$ mice was pro-cessed using KAPA RNA Hyperprep Kit with RiboErase (Kapa Biosystems/Roche) because of the discontinuation of the Lex-ogen library construction kit. Libraries were sequenced with single-end 75-nt reads (8-wk-old $Prpf8^{\Delta 17}$ mice and 12-wk-old $Prpf8^{Y2334N}$ animals) or with paired-end 75 + 75-nt reads (4-wk-old $Prpf8^{\Delta 17}$ mice and 4-wk-old $Prpf8^{Y2334N}$ animals) on Illumina NextSeq 500 System. A total of 10–14 animals of both genders were selected for indi-vidual RNA-Seq runs; representation of individual genotypes was thereby as follows: 12-wk-old $Prpf8^{Y2334N}$ animals (a total of 14 animals): n = 6 for $Prpf8^{Y2334N/Y2334N}$ biological replicates (3x male, 3x female), n = 4 for $Prpf8^{Y2334N/wt}$ mice (2xmale, 2xfemale), and n = 4 for $Prpf8^{wt/wt}$ litter-matched controls (2xmale, 2xfemale); 8-wk-old $Prpf8^{\Delta 17}$ mice (a total of 10 animals): n = 6 for $Prpf8^{\Delta 17/\Delta 17}$ biological replicates (4xmale, 2xfemale) and n = 4 for $Prpf8^{wt/wt}$ litter-matched controls (2xmale, 2xfemale); 4-wk-old $Prpf8^{\Delta 17}$ animals (a total of 10 animals): n = 6 for $Prpf8^{\Delta 17/\Delta 17}$ biological replicates (4xmale, 2xfe-male) and n = 4 for $Prpf8^{wt/wt}$ litter-matched controls (2xmale, 2xfemale); and 4-wk-old $Prpf8^{Y2334N}$ animals (a total of 10 animals): n = 6 for $Prpf8^{Y2334N/Y2334N}$ biological replicates (4xmale, 2xfemale) and n = 4 for $Prpf8^{wt/wt}$ litter-matched controls (2xmale, 2xfemale).

### Differential gene expression and alternative splicing analyses

For gene-level expression quantification, a bioinformatic pipeline nf-core/rnaseq (Ewels et al, 2020) was used (version 1.3 for 8-wk-old $Prpf8^{\Delta 17}$ mice and 12-wk-old $Prpf8^{Y2334N}$ animals; version 1.4.2 for 4-wk-old $Prpf8^{\Delta 17}$ mice and 4-wk-old $Prpf8^{Y2334N}$ animals). Individual steps included removing sequencing adaptors with Trim Galore! (www.bioinformatics.babraham.ac.uk/projects/trim_galore), map-ping to reference genome GRCm38 (Ensembl annotation version 94) with HISAT2 (Kim et al, 2015), and quantifying expression on the gene level with featureCounts (Liao et al, 2014). Per gene uniquely mapped read counts served as input for differential expression analysis using the DESeq2 R/Bioconductor package (Love et al, 2014), done separately for each sequencing run data. Before the analysis, genes not detected in at least two samples were

discarded. We supplied an experimental model assuming sample genotype as a main effect, while accounting for breeding litter (for 8-wk-old $Prpf8^{\Delta 17}$ mouse samples) or gender (for 4-wk-old $Prpf8^{\Delta 17}$ mice and 4-wk-old $Prpf8^{Y2334N}$ animals) as a batch effect. Resulting per gene expression $\log_2$ fold changes shrunken using the adaptive shrinkage estimator (Stephens, 2017) were used for differential expression analysis. Genes exhibiting |$\log_2$ fold change| > 1 and statistical significance FDR < 0.05 between compared groups of samples were considered as differentially expressed. Next, gene set overrepresentation analysis with differentially expressed genes was done using the gene length bias–aware algorithm imple-mented in the goseq R/Bioconductor package (Young et al, 2010) against KEGG pathways and GO term gene sets. Alternative splicing analysis was assessed using the ASpli R/Bioconductor package (Mancini et al, 2021), version 2.0.

## circRNA quantification and differential expression

circRNA back-spliced junction sites were determined indepen-dently in each individual sequencing run data using the CIRI2 tool (Gao et al, 2018) following the respective guidelines with the same genome reference data supplied for the gene-level analysis. The expression of circRNAs was consequently quantified using the CIRI2 (adaptor-trimmed data for 8-wk-old $Prpf8^{\Delta 17}$ mice and 12-wk-old $Prpf8^{Y2334N}$ animals) or the CIRIquant (Zhang et al, 2020) (full-length data for 4-wk-old $Prpf8^{\Delta 17}$ mice and 4-wk-old $Prpf8^{Y2334N}$ animals) tool. Analysis of alternative splicing and quantification of exon expression within detected circRNAs were done for 4-wk-old $Prpf8^{\Delta 17}$ mice and 4-wk-old $Prpf8^{Y2334N}$ animals with the CIRI-AS tool provided alongside CIRI2. Differential expression analysis of circRNAs was done using either the DESeq R/Bioconductor package (for 8-wk-old $Prpf8^{\Delta 17}$ mice and 12-wk-old $Prpf8^{Y2334N}$ animals) or the edgeR (McCarthy et al, 2012) R/Bioconductor package pipeline provided alongside CIRIquant (for 4-wk-old $Prpf8^{\Delta 17}$ mice and 4-wk-old $Prpf8^{Y2334N}$ animals). For all sequencing data, the same parameters for defining differentially expressed circRNAs were set (|$\log_2$ fold change| > 1 and FDR < 0.05). For parental genes of differentially expressed circRNAs, GO term gene set overrepresentation was analyzed (for 8-wk-old $Prpf8^{\Delta 17}$ mice and 12-wk-old $Prpf8^{Y2334N}$ animals only) using the goseq R/Bioconductor package without gene length bias considered.

## Generation and analysis of circRNA reporters

The minimal sequences surrounding the circularized exons of the differentially expressed circRNAs were identified from the RNA-Seq data and were sorted keeping the maximum length of the construct within 5 kbp and simultaneously taking into account the repeat elements important for back-splicing on both sides of the back-spliced junction. The final reporter construct constituted three exons (two exons getting back-spliced and a downstream exon to denote the linear splicing with the middle exon) and two introns between them. Classical restriction cloning (Kdm2a, restriction sites NheI and BamHI) and Gibson assembly technique (for Kmt2a and Rad52) were used to create circRNA expression vectors of the corresponding genes Kmt2a (chr9:44,828,270–44,830,122), Kdm2a (chr19:4,319,095–4,322,547), and Rad52 (chr6:119,919,824–119,922,533). Genomic DNA was isolated from mouse embryonic fibroblast (MEF)

cells using the QIAamp DNA mini kit (QIAGEN), PCR-amplified, and subsequently cloned in vector pEGFPC1 with a constitutive promoter. Positive clones were confirmed by sequencing.

Human cervical adenocarcinoma-derived epithelial cell line HeLa (ATCC) and immortalized hTERT-RPE cells expressing WT-PRPF8-GFP (RPE$^{wt/wt}$) and Y2334N-PRPF8-GFP (RPE$^{N/N}$) were maintained in DMEM (Thermo Fisher Scientific), supplemented with 10% FBS (Thermo Fisher Scientific) and penicillin–streptomycin mix (Thermo Fisher Scientific). Cells were transfected using Lipofectamine LTX Reagent or 3000 (Thermo Fisher Scientific) according to the manufacturer's protocol, and analyzed 24 h post-plasmid delivery.

Total RNA was isolated by TRIzol (Ambion) and treated with 1 $\mu$l of Turbo DNase (2 U/$\mu$l; Thermo Fisher Scientific). 200 ng of RNA was used for cDNA production by reverse transcription (Superscript III; Thermo Fisher Scientific) using random hexamers, and 1/10 of the cDNA was subsequently used for quantitative PCR. One set of primers was designed in reverse orientation to span the back-spliced junction between the 5′ end of exon 1 and the 3′ end of exon 2. The circular form of RNA was confirmed by resistance to the RNase R (McLab) treatment (10 min at 37°C) before RT–qPCR. The second set of primers span the junction between the 3′ end of exon 2 and the 5′ end of exon 3. This amplifies the linear spliced variant. The fold change expression level ($2^{-\Delta\Delta Ct}$) was normalized to the negative control (pEGFPC1) and to the housekeeping gene (GAPDH) and presented as a ratio of circular to linear RNA between the WT and mutant cells.

## Data Availability

The datasets generated and/or analyzed during the current study are available in the following repositories:

- RNA-Seq data: Array Express accession E-MTAB-10753
- RT–qPCR linear model (github.com/cas-bioinf/prpf8).

## Supplementary Information

## Acknowledgements

The authors thank Jana Machatova-Krizova, Sarka Kocourkova, and Martina Krausova for excellent technical assistance; Gabriela Vavrova for her careful approach to mouse keeping; Peter Makovicky for his significant contribution to the histopathological analyses; and Dr. Ondrej Machon for assistance with animal organ preparation. We would also like to express our gratitude to Assoc.Prof. Jakub Otahal from the Institute of Physiology of the Czech Academy of Sciences for his help with assignment of stereotaxic coordinates used for the description of histopathological sections. We would also like to thank Dr. Trevor Epp from the Institute of Molecular Genetics of the Czech Academy of Sciences for helpful discussion on the transmission ratio distortion. This work was supported by the Czech Academy of Sciences RVO 68378050 and 68378050-KAV-NPUI and the Czech Science Foundation (20-04099S). M Krausová was supported by the Czech Academy of Sciences (L200521652) and the Czech Academy of Sciences/Deutscher Akademischer Austauschdienst Mobility Programme (DAAD-17-10). The Czech Centre for Phenogenomics (to D Zudová, J Lindovský, A Kubik-Zahorodna, M Pálková, J Procházka, and R Sedláček) was supported by the project LM2018126 Czech Centre for Phenogenomics provided by MEYS. The additional funding for CCP was provided by projects CZ.02.1.01/0.0/0.0/16_013/0001789 and CZ.02.1.01/0.0/0.0/18_046/0015861 provided by MEYS and ESIF. This work was also supported by the ELIXIR CZ research infrastructure project (MEYS Grant No: LM2018131) including access to computing and storage facilities (to M Modrák, J Kubovčiak, and M Kolář). The microscopy images were acquired at the Light Microscopy Core Facility, Institute of Molecular Genetics in Prague, Czech Republic, supported by MEYS (LM2015062 and CZ.02.1.01/0.0/0.0/16_013/0001775) and OPPK (CZ.2.16/3.1.00/21547).

## Author Contributions

M Krausová: conceptualization, formal analysis, supervision, funding acquisition, validation, investigation, visualization, methodology, and writing—original draft.

M Kreplová: validation, investigation, and methodology.

P Banik: validation, investigation, and methodology.

Z Cvačková: resources, formal analysis, and validation.

J Kubovčiak: formal analysis.

M Modrák: software and formal analysis.

D Zudová: investigation and methodology.

J Lindovský: investigation and methodology.

A Kubik-Zahoradna: investigation and methodology.

M Pálková: investigation and methodology.

M Kolář: supervision and funding acquisition.

J Procházka: supervision, investigation, and project administration.

R Sedláček: supervision and funding acquisition.

D Staněk: conceptualization, resources, supervision, funding acquisition, project administration, and writing—original draft, review, and editing.

## Declarations

### Ethical approval

All animal procedures were conducted following the European Union guidelines (regulation n°86/609) and the Czech law regulating animal experimentation (246/1992 Sb.), and approved by the Ethics Committee of Czech Republic (reference number 31255/2019-MZE-18134, file ID 16OZ9707/2019-18134, valid till July 2, 2024).

### Conflict of Interest Statement

The authors declare that they have no conflict of interest.

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
