## [Reviewer comments · Life Science Alliance]

Life Science Alliance

Retinitis pigmentosa associated mutations in Prpf8 cause degeneration of cerebellar granule cells

Michaela Krausova, Michaela Kreplová, Poulami Banik, Zuzana Cvačková, Jan Kubovciak, Martin Modrak, Dagmar Zudova, Jiri Lindovsky, Agnieszka Kubik-Zahoradna, Marcela Palkova, Michal Kolar, Jan Prochazka, Radislav Sedlacek, and David Stanek
DOI: <https://doi.org/10.26508/lsa.202201855>

Corresponding author(s): David Stanek, Czech Academy of Sciences, Institute of Molecular Genetics

Review Timeline:

Submission Date:	2022-11-25
Editorial Decision:	2023-01-04
Revision Received:	2023-02-23
Editorial Decision:	2023-03-15
Revision Received:	2023-03-21
Accepted:	2023-03-21

Scientific Editor: Novella Guidi

Transaction Report:

January 4, 2023

Re: Life Science Alliance manuscript #LSA-2022-01855

Dr. David Stanek
Czech Academy of Sciences, Institute of Molecular Genetics
Department of RNA Biology
Videnska 1083
Prague 14220
Czech Republic

Dear Dr. Stanek,

Thank you for submitting your manuscript entitled "Retinitis pigmentosa associated mutations in Prpf8 cause degeneration of cerebellar granule neurons" to Life Science Alliance. The manuscript was assessed by expert reviewers, whose comments are appended to this letter. We invite you to submit a revised manuscript addressing the Reviewer comments.

Thank you for this interesting contribution to Life Science Alliance. We are looking forward to receiving your revised manuscript.

Sincerely,

B. MANUSCRIPT ORGANIZATION AND FORMATTING:

Reviewer #1 (Comments to the Authors (Required)):

Krausova et al. generated mouse models to study the effects of selected mutations in the gene encoding core splicing factor PRPF8, which have been linked to retinitis pigmentosa (RP) in humans. Specifically, they generated animals that heterozygously or homozygously produced a PRPF8 variant, in which the penultimate Y2334 was converted to an N, or a variant with an extended C-terminus. While the former variant corresponded to an authentic RP-linked PRPF8 variant in humans, the latter variant mimicked, but did not precisely correspond to, certain RP-linked human PRPF8 variants. Still, as diverse C-terminally extended PRPF8 variants have been linked to RP, it is also a suitable RP model protein. The authors then assessed phenotypes in the mutated animals and studied the postnatal maturation of the cerebellar tissue, documenting the timeline of neuronal degeneration. They then used RNA-seq to monitor changes in the transcriptomes of cerebellar tissue before and after the onset of the degeneration. Significant gene expression changes were observed only in homozygous animals, some of which, based on known functions of affected genes, may be functionally related to the neuronal degeneration. Changes in alternative splicing could not be unequivocally related to the PRPF8 variants due to the mixture of cell types analyzed. However, a fairly large number of circular RNAs were up- or downregulated in the mutant animals. The authors also made the interesting observation that PRPF8 protein levels are predominantly regulated on the protein level, not the transcript level.

This study relates to the important question of how mutations in ubiquitously expressed genes can lead to tissue-specific pathological situations. In this respect, the authors present an interesting hypothesis consistent with their data - the natural decay of splicing factors observed in certain neuronal tissues/cell types (which coincides with the time of the onset of neuronal degeneration in RP-model animals, and which is exacerbated in the presence of RP-linked mutations in the *prpf8* gene) sensitizes the corresponding cells, such that RP-linked mutations result in tissue-specific effects.

Overall, the approaches and results are clearly presented. The work conducted appears to be technically sound. While the study does not rigorously link molecular mechanisms to disease phenotypes, in part due to our poor understanding of the functions of many circRNAs, it provides a number of interesting observations. This reviewer has only few specific comments that the authors may want to consider in a revised version of their manuscript:

1. The conclusion, based on pulldown assays, that the PRPF8 variants investigated do not exhibit altered interactions with other spliceosomal factors tested, in particular with SNRNP200, is not entirely justified. Specifically, the region of PRPF8 affected by the mutations contributes only very little to the binding affinity of PRPF8 to SNRNP200, but it represents an important regulatory element for the SNRNP200 helicase. The mutations affect an intrinsically disordered C-terminal region of PRPF8 that can be inserted into the SNRNP200 RNA-binding channel and thereby inhibit the helicase. While the overall affinity to SNRNP200 may not be affected, the mutations may well influence if, and how efficiently, the C-terminal region of PRPF8 can insert into SNRNP200 and regulate its activity.
2. Somewhat related to the above point, it would be nice if in the Discussion the authors could try to also establish a link to the molecular organization and interactions of PRPF8 in the spliceosome, which by now have been imaged at the atomic level in many studies. Can they speculate how the mutations that affect a very specific region of PRPF8 could lead to mis-splicing of mRNAs or aberrant production of circRNAs? For example, can the authors suggest an explanation for the observation that the altered circRNA expression correlates with a suboptimal 3'-splice site of the alternative exons involved; how could the investigated PRPF8 variants lead to improper recognition of these 3'-splice sites? Furthermore, although the same region of PRPF8 is affected by the two investigated RP-linked mutations, very different sets of pre-mRNA splicing events were affected by the two mutations - how could this be understood based on the known/presumed functions of the affected PRPF8 region? Could the two mutations affect the function of this region in different ways?
3. The authors observed gene expression effects only in homozygous animals, while humans suffer from RP if only one *prpf8* allele is affected. Can the authors suggest an explanation for this apparent inconsistency?

Reviewer #2 (Comments to the Authors (Required)):

The paper by Krausova investigates mutations relevant for retinitis pigmentosa (RP), a rare, but clinically highly relevant

progressive degenerative disease.

The authors developed new mouse lines (CRISPR/Cas9) in which they mimicked disease-related mutations in one of the master genes (PRPF8) associated with RP. Using CRISPR/Cas9 explains why the engineered mutations are not directly mimicking mutations found in humans.

Anyway, the aim of the paper is not to mimic RP, but to develop mouse lines that help to better understand how PRPF8 acts on a molecular and cellular level. This is indeed important as these mouse lines now offer new insights in how PRPF8 can affect neurons. Furthermore, the animals are viable and can be bred to homozygosity.

In summary, I'm impressed by the large amount of data, the excellent description and phenotyping of the mouse models, and the meaningful results. In my opinion, there is not much to add, the data is already overwhelming (and not easy to review in depth in a reasonable time).

Importantly, there is much new information about the function of PRPF8 with respect to neurodegeneration. Surprising are the observed progressive neurodegenerative changes in the cerebellum of both mouse lines. Furthermore, from a general perspective, we also learn so much about how granule cell loss affects the cerebellum and that this does not affect other granule cells in other brain areas (e.g. hippocampus - as expected) (supplemental material - super interesting!).

There are just some minor points from my side:

1) The authors use the term granule neuron as well as granule cell. I recommend to stick to the traditional term granule cell (cerebellar granule cell).

2) The authors should guide the readers (maybe in the abstract) to the overall importance regarding granule cell degeneration. I'm sure that many experts from neurobiology would be interested in the observations.

3) Page 6 - "exhibited tremor indicating a potential neurodegeneration"

How was this observed and what does tremor mean here (limbs, observed during walking, There might be a little video for the supplements).

4) page 6: mice had to be euthanized

Why? According to which criterion (with respect to animal scoring according to the animal protection rules - Tremor?, loss of weight? Suffering?)

5) The authors might think about the term 'activated microglia'. A rather new perspective article in Neuron asks to adapt the term to our current knowledge (DOI: 10.1016/j.neuron.2022.10.020).

6) page 10: Surprisingly, our data show that the regulation of splicing protein occurs at the protein level because we observed, that the dosage compensation for the missing Prpf8 allele occur at the protein but not mRNA level (Fig. 8).

This sentence is not fully covered by data. The statement should be weakened. (suggest / indicate / might)

7) The discussion is very much focused on spliceosome biology, but the main finding, neurodegeneration of granule neurons, while other cerebellar elements survive, is largely ignored. A limitation section is fully missing (e.g. mutational mimic, understanding of neurodegeneration, is the effect cell-autonomous or systemic).

8) Method section:

RRIDs (antibodies, software, antibody documentation (dilution from original stock) are missing.

9) How are the mice available to other researchers?

Randomly observed typos:

Page 10: '4 weeks old animals' (4 week old - common typo in the manuscript)

Page 11: 'linear-to-circular ration' (ratio)

Reviewer #3 (Comments to the Authors (Required)):

The major goal of the study by Krausova et al. was to provide novel insights into cellular and molecular mechanisms underlying progressive cell-type specific neurodegeneration caused by the malfunction of a critical splicing factor Prpf8. The authors generated mouse models bearing three different Prpf8 mutant alleles, Prpf8Y2334N, Prpf8Δ17, and Prpf8Δ366. While Prpf8Δ366 represents a null, loss-of-function allele, Prpf8Y2334N and Prpf8Δ17 resemble mutations detected in retinitis pigmentosa patients. In contrast to the embryonic lethality of homozygous Prpf8Δ366 animals, Prpf8 Y2334N/Y2334N and Prpf8Δ17/Δ17 animals were viable, manifesting signs of neurodegeneration and marked neuronal damage by 15-weeks of age.

The authors performed a careful, rigorous, and detailed analysis of the two Prpf8 homozygous mutant strains at different time points before the manifestation of severe locomotion impairment, revealing neuronal cell death and reduction of granule neuron density. In contrast, glial cell populations were dramatically increased, indicative of neuroinflammation. Intriguingly, the author identified the posterior part of the cerebellar cortex, rather than retina, as the most affected already at 6-weeks of age.

Based on the co-IP experiments, the authors concluded that similar to Prpf8Y2334N (Malinova et al., 2017), the C-terminal aberrant extension of Prpf8Δ17 protein did not affect interactions with a selected set of known splicing factors (Prpf6 and

Snrnp200) favoring hypothesis of altered splice site selection rather than overall deficiency of functional spliceosomes.

Transcriptome profiles obtained from cerebella of 12-weeks-old Prpf8 Y2334N/Y2334N and 8-weeks-old Prpf8 Δ 17/ Δ 17 animals and wild-type aged matched littermates supported the observed phenotypes highlighting gain of glia-specific gene signature at the expense of granular neurons and synapse organization and function. The authors identified differential expression of several circRNAs in Prpf8 mutant cerebella already at 4 weeks of age compared to control. The impact of Prpf8 Δ 17 on backsplicing was further demonstrated on one of the three tested reporter constructs in cultured RPE cells.

Analysis of alternative splicing revealed alterations in all types of splicing events. However, the changes were unique to the two Prpf8 mutations and did not show enrichment for any functional categories.

Finally, the authors demonstrated a dramatic decline of selected U5 snRNP spliceosome components on a protein but not a transcript level, starting from week 2 postnatally. Again the drop was the most prominent in neuronal tissue of the cerebellum and retina.

The authors concluded that the rapid decline of the key U5 snRNP spliceosome components, including Prpf8, upon birth sensitizes neuronal tissue of the cerebellum to the dysregulation of circRNA expression caused by RP-Prpf8 mutations. The study further supports the notion that neurodegeneration is not caused by Prpf8 haploinsufficiency.

This is a rigorous study presenting phenotypic and molecular description of the new murine adRP disease models, providing a correlative link between Prpf8 protein malfunction and deregulation of circRNAs and loss of a specific neuronal population.

Major concerns:

The focus on circRNAs is logical given that their formation is directly linked to splicing, they are abundant and enriched in the neuronal tissues, and a growing body of functional evidence implicates circRNA in controlling neuronal development and function, including cell survival and the establishment and/or maintenance of synapses.

My primary concern is the specificity of circRNA misexpression. The changes in circRNAs have been detected as significant in the initial dataset generated from 12-weeks-old Prpf8 Y2334N/Y2334N and 8-weeks-old Prpf8 Δ 17/ Δ 17 animals. Given the extent of gene deregulation, would one not expect circRNAs to be changed as well? In other words, is the deregulation of circRNAs specific to the presence of Prpf8 mutations, or would it be observed in conditions where gene expression is deregulated to a similar extent by other insults? Did authors detect Prpf8 mutant-dependent bias in other transcript biotypes?

No significant differences in circRNAs were detected in the 4-week RNA-seq dataset. Still re-analysis by RT-qPCR showed shifts in major circRNAs in the direction observed at a later timepoint. Can authors comment on how they controlled for the specificity of circRNA detection? Did the authors use RNase R treatment to avoid potential artifacts caused by the template switching?

Using immunohistochemistry, the authors corroborated the key differences/changes revealed by the transcriptome. They have done an excellent job of describing the individual panels allowing easy comparison. The majority of images are convincing, with differences being clearly visible. The possible exceptions are panels dedicated to Rbfox3 (NeuN) and synaptic marker PSD95 (Dlg4). The insets from control brains would be helpful for comparison to appreciate the decrease in levels Supplementary Figure S9 A, B).

Figure 6 and 7, I am not convinced about the claim that the drop in Prpf8 protein is more dramatic in the cerebella of Prpf8Y2334N/Y2334N mice relative to control mice. Running all samples together on the same blot would help to provide stronger evidence.

Could authors explain or speculate about a Prpf6 double band detected specifically in liver samples compared to a single band in brain and retina tissue?

Minor comments:

Writing

Figure legend 6-8, should read "in the same tissue and then calculated..."

Please revise the following sentences:

"The fold change expression level ($2^{-\Delta\Delta Ct}$) was normalized to normalizing to the negative control (pEGFPC1) and to housekeeping gene (GAPDH) and presented as a ratio of circular/ linear RNA between the wild type and mutant cells. "Normalizing to" does not seem to fit.

"Total RNA were isolated by Trizol (Ambion) and treated with DNase. 200 ng of RNA was used for complementary DNA (cDNA) production by reverse transcription (Superscript III, ThermoFischer Scientific) using random hexamers and 1/10th dilution of the cDNA." The authors might want to write "DNase I" and include the source. For what the authors used the 1/10th dilution of the cDNA during cDNA synthesis?

The authors mention using HeLa cells in the M&M section describing the procedure for testing the impact of Prpf8 on back splicing and production of circRNAs. I could find only data for RPE in the results. Weren't HeLa cells used to test the efficacy of the gRNAs rather than backsplising?

The authors might want to double-check referencing the Supplementary data, Tables in particular, throughout the manuscript, including the Figure legends. For example, in the Figure legend S11, Supplementary Table 7 does not contain the data to which the authors refer to. It should be Supplementary Table 2.

The authors show that in heterozygous Prpf8 Δ 366/+ animals the Prpf8 protein level reaches that of wild type littermates despite lower transcription. Do Prpf8 Δ 366/+ show any precocious signs of aging and/or aging-associated neurodegeneration? Can authors speculate or comment on whether heterozygosity becomes a limiting factor during aging? Does Prpf8 protein in brains of 4-weeks old Prpf8 Δ 366/+ drop so dramatically as in controls?

The observed postnatal regulation of U5 snRNP proteins, including Prpf8, but not their mRNAs is exciting. It is intriguing how mouse cerebellum/retina can function with about 15 % or lower levels of the original amount of Prpf8 protein from a few weeks postnatally. A similar regulatory phenomenon has been observed by Stankovic et al 2020. The authors noted that despite overexpressing wild-type or RP-Prp8 mutant variants in the developing Drosophila eye using a strong ey-Gal4 driver, the total amount of Prp8 protein did not exceed that of control, implying a regulation at the protein rather than mRNA level.

Reviewer #1 (Comments to the Authors (Required)):

We thank the reviewer for positive comments on our manuscript as well as helpful suggestion that improved the text.

1. The conclusion, based on pulldown assays, that the PRPF8 variants investigated do not exhibit altered interactions with other spliceosomal factors tested, in particular with SNRNP200, is not entirely justified. Specifically, the region of PRPF8 affected by the mutations contributes only very little to the binding affinity of PRPF8 to SNRNP200, but it represents an important regulatory element for the SNRNP200 helicase. The mutations affect an intrinsically disordered C-terminal region of PRPF8 that can be inserted into the SNRNP200 RNA-binding channel and thereby inhibit the helicase. While the overall affinity to SNRNP200 may not be affected, the mutations may well influence if, and how efficiently, the C-terminal region of PRPF8 can insert into SNRNP200 and regulate its activity.

The reviewer is correct that the Y2334N substitution in Prpf8 studied in this manuscript should not considerably affect interaction with Snrnp200. Consistently, we previously showed that the same substitution in human PRPF8 does not reduce U5 snRNP assembly and interaction with SNRNP200 (Malinova et al 2017, DOI: 10.1083/jcb.201701165). However, these experiments were performed in human cell culture and we wanted to confirm that this also applies to the proteins in mouse cerebellum. In addition, the second mouse strain (Prpf8^{Δ17}), which we created in the course of this project, has never been tested for interaction with other U5 snRNP proteins. We therefore performed immunoprecipitation of both mutated proteins: 1. To confirm previous findings done on human cells with PRPF8^{Y24334N} protein (Prpf8^{Y2334N} allele) and 2. To probe the interaction of a new Prpf8 variant with protein markers of U5 snRNP (Prpf8^{Δ17} allele). We believe that our immunoprecipitation experiments allow us to draw the conclusion that neither of these two tested mutations significantly affect interaction with Snrnp200 and Prpf6 and in general with U5 snRNP. We modified the text in Discussion (p. 16, bottom of the page) to communicate and discuss our results more thoroughly. For the potential effect of the mutations on prpf8 and Snrnp200 function, please see the comments to your next point.

2. Somewhat related to the above point, it would be nice if in the Discussion the authors could try to also establish a link to the molecular organization and interactions of PRPF8 in the spliceosome, which by now have been imaged at the atomic level in many studies. Can they speculate how the mutations that affect a very specific region of PRPF8 could lead to mis-splicing of mRNAs or aberrant production of circRNAs? For example, can the authors suggest an explanation for the observation that the altered circRNA expression correlates with a suboptimal 3'-splice site of the alternative exons involved; how could the investigated PRPF8 variants lead to improper recognition of these 3'-splice sites? Furthermore, although the same region of PRPF8 is affected by the two investigated RP-linked mutations, very different sets of pre-mRNA splicing events were affected by the two mutations - how could this be understood based on the known/presumed functions of the affected PRPF8 region? Could the two mutations affect the function of this region in different ways?

When the C-terminal tail of PRPF8 is inserted in the RNA channel of SNRNP200, the substitutions Y2334N in PRPF8 and S1087L and R1090L in SNRNP200 are located

in a close vicinity (Mozaffari-Jovin 2013, DOI: 10.1126/science.1237515). Similarly to Y2334N in PRPF8, both mutations in SNRNP200 do not alter interaction of SNRNP200 with other snRNP proteins (Cvackova et al. 2014, DOI: 10.1002/humu.22481) and apparently these three substitutions have different mode of action than most other RP mutations in snRNP-specific proteins that lead to snRNP destabilization. The most probable mode of actions of these three mutations (Y2334N in Prpf8 and S1087L and R1090L in SNRNP200) is deregulation of the helicase activity of SNRNP200, where the C-terminus of PRPF8 has a critical role. However, how this deregulation translates into changes at specific splice sites is currently unclear and answer to this question requires additional biochemical and structural studies to fully uncover the molecular mechanism. We added a speculation about deregulation of SNRNP200 helicase activity caused by the studied RP mutations into Discussion (p. 16, bottom paragraph and p. 17 top paragraph).

3. The authors observed gene expression effects only in homozygous animals, while humans suffer from RP if only one prpf8 allele is affected. Can the authors suggest an explanation for this apparent inconsistency?

We significantly rewrote the first four paragraphs of Discussion, where we discuss previous findings using mouse models of RP and propose hypotheses that could explain the difference between humans and mice retina and cerebellum (pp. 16-17).

Reviewer #2 (Comments to the Authors (Required)):

We thank the reviewer for supporting comments on our manuscript and suggestions that helped to improve our work.

There are just some minor points from my side:

1) The authors use the term granule neuron as well as granule cell. I recommend to stick to the traditional term granule cell (cerebellar granule cell).

We changed the text and use now the term "granule cell" throughout the whole paper including the title. Thank you!

2) The authors should guide the readers (maybe in the abstract) to the overall importance regarding granule cell degeneration. I'm sure that many experts from neurobiology would be interested in the observations.

We changed the Abstract to state that specifically granule cells in cerebellum are sensitive to mutations in Prpf8.

3) Page 6 - "exhibited tremor indicating a potential neurodegeneration"

How was this observed and what does tremor mean here (limbs, observed during walking, There might be a little video for the supplements).

We added a video of an affected mouse exhibiting the tremor (Supplementary video 1).

4) page 6: mice had to be euthanized

Why? According to which criterion (with respect to animal scoring according to the animal protection rules - Tremor?, loss of weight? Suffering?)

The heterozygotic animals did not develop any harmful phenotype and the mice strains were kept in heterozygotic state. For the experiments, the homozygous animals were mostly analyzed before week 15 when the tremor and locomotion disturbances started. For a complete analysis of the phenotype, we had to keep a few animals longer (up to 22 weeks), when they developed ataxia and that was the reason why they were euthanized. We state this fact in the revised text (p. 6, bottom of the page).

5) The authors might think about the term 'activated microglia'. A rather new perspective article in Neuron asks to adapt the term to our current knowledge (DOI: 10.1016/j.neuron.2022.10.020).

In the revised version, we avoided the term "activated microglia" and rather write about microglia that express markers found previously in disease- and neurodegeneration-associated microglia (p. 9, top of the page). We also remove the term from Supplementary figure and the figure legend. Thank you!

6) page 10: Surprisingly, our data show that the regulation of splicing protein occurs at the protein level because we observed, that the dosage compensation for the missing Prpf8 allele occur at the protein but not mRNA level (Fig. 8).

This sentence is not fully covered by data. The statement should be weakened. (suggest / indicate / might)

We softened the statement in Discussion as suggested (p. 17, middle of the page).

7) The discussion is very much focused on spliceosome biology, but the main finding, neurodegeneration of granule neurons, while other cerebellar elements

survive, is largely ignored. A limitation section is fully missing (e.g. mutational mimic, understanding of neurodegeneration, is the effect cell-autonomous or systemic).
We significantly rewrote first four paragraphs of Discussion to address the points raised by the reviewer. However, I have to admit that our major expertise lays in RNA splicing biology and therefore I do not feel fully competent to discuss our findings in details of a complex cerebral development and neurodegeneration. The animals are available for research community and experts in the neurodegenerative field can further analyze them.

8) Method section:

RRIDs (antibodies, software, antibody documentation (dilution from original stock) are missing.

All the information was included into revised version of the manuscript.

9) How are the mice available to other researchers?

We keep breeding the animals and they will be freely available for the academic research once our work is published.

Randomly observed typos:

Page 10: '4 weeks old animals' (4 week old - common typo in the manuscript)

Corrected, thank you!

Page 11: 'linear-to-circular ration' (ratio)

Corrected, thank you!

Reviewer #3 (Comments to the Authors (Required)):

We thank the reviewer for positive comments and suggestions that improved our manuscript.

Major concerns:

My primary concern is the specificity of circRNA misexpression. The changes in circRNAs have been detected as significant in the initial dataset generated from 12-weeks-old Prpf8 Y2334N/Y2334N and 8-weeks-old Prpf8Δ17/Δ17 animals. Given the extent of gene deregulation, would one not expect circRNAs to be changed as well? In other words, is the deregulation of circRNAs specific to the presence of Prpf8 mutations, or would it be observed in conditions where gene expression is deregulated to a similar extent by other insults? Did authors detect Prpf8 mutant-dependent bias in other transcript biotypes?

We analyzed further potential defects in splicing that involved intron retention, exon skipping, and alternative 5′-/3′-splice site recognition. In both mouse strains, we observed numerous changes in intron retention but we were not able to identify any gene, where the intron retention would be same or similar between both mouse strains. Similarly, comparison of exon skipping and alternative 5′/3′ splice site usage did not reveal any overlap between the two mouse strains. These results are presented at p. 12, middle paragraph and Tables S8 and S9.

No significant differences in circRNAs were detected in the 4-week RNA-seq dataset. Still re-analysis by RT-qPCR showed shifts in major circRNAs in the direction observed at a later timepoint. Can authors comment on how they controlled for the specificity of circRNA detection? Did the authors use RNase R treatment to avoid potential artifacts caused by the template switching?

In the particular experiment mentioned by the reviewer (Fig. 5) we did not use RNA R treatment because we aimed to determine ratio between circular RNAs and linear mRNAs expressed from the same gene. However, the specificity of the primers for circular RNA has been done prior the experiment and the RNA R treatment is described in Material and Methods (p. 35, middle paragraph).

Using immunohistochemistry, the authors corroborated the key differences/changes revealed by the transcriptome. They have done an excellent job of describing the individual panels allowing easy comparison. The majority of images are convincing, with differences being clearly visible. The possible exceptions are panels dedicated to Rbfox3 (NeuN) and synaptic marker PSD95 (Dlg4). The insets from control brains would be helpful for comparison to appreciate the decrease in levels Supplementary Figure S9 A, B).

Done, we included the required insets from WT brains in the revised Figs S9A and B.

Figure 6 and 7, I am not convinced about the claim that the drop in Prpf8 protein is more dramatic in the cerebella of Prpf8Y2334N/Y2334N mice relative to control mice. Running all samples together on the same blot would help to provide stronger evidence.

Here, our conclusion is based on quantification of the signal with respect to house keeping genes and comparison of the graphs shown in Figs. 6 and 7. In the revised version of the manuscript, we further run samples from cerebellum of WT and

Y2334N animals on the same gel as suggested by the reviewer and these results are now shown at new Fig. S15A.

Could authors explain or speculate about a Prpf6 double band detected specifically in liver samples compared to a single band in brain and retina tissue?

We did not find any alternative transcripts of mouse Prpf6 annotated in Ensemble and USCS Genome browsers, which suggests that alternative promoters or splicing do not generate alternative forms of Prpf6. However, Prpf6 is phosphorylated at several residues (www.uniprot.org). The phosphorylation pattern might differ among different tissues, which may result in the observed double band in the liver sample.

Minor comments:

Writing

Figure legend 6-8, should read "in the same tissue and then calculated..."

Corrected, thank you!

Please revise the following sentences:

"The fold change expression level ($2^{-\Delta\Delta Ct}$) was normalized to normalizing to the negative control (pEGFPC1) and to housekeeping gene (GAPDH) and presented as a ratio of circular/ linear RNA between the wild type and mutant cells. "Normalizing to" does not seem to fit.

Done.

"Total RNA were isolated by Trizol (Ambion) and treated with DNase. 200 ng of RNA was used for complementary DNA (cDNA) production by reverse transcription (Superscript III, ThermoFischer Scientific) using random hexamers and 1/10th dilution of the cDNA." The authors might want to write "DNase I" and include the source. For what the authors used the 1/10th dilution of the cDNA during cDNA synthesis?

1/10th dilution of the cDNA was used quantitative PCR - we explained it in the text (p. 35 - middle paragraph). We also added the name and the source of the used DNase. Thank you!

The authors mention using HeLa cells in the M&M section describing the procedure for testing the impact of Prpf8 on back splicing and production of circRNAs. I could find only data for RPE in the results. Weren't HeLa cells used to test the efficacy of the gRNAs rather than backsplising?

Yes, the reviewer is correct, HeLa cells were used to test the efficacy of the gRNAs. We added a new section in Material and methods describing editing of RPE cells (p. 23) and a new Fig. S17 where we provide basic characterization of these cells.

The authors might want to double-check referencing the Supplementary data, Tables in particular, throughout the manuscript, including the Figure legends. For example, in the Figure legend S11, Supplementary Table 7 does not contain the data to which the authors refer to. It should be Supplementary Table 2

Thank you for pointing this out, we went through the text and figure legend referencing to the Supplementary data and corrected mistakes.

The authors show that in heterozygous Prpf8 Δ 366/+ animals the Prpf8 protein level reaches that of wild type littermates despite lower transcription. Do Prpf8 Δ 366/+

show any precocious signs of aging and/or aging-associated neurodegeneration? Can authors speculate or comment on whether heterozygosity becomes a limiting factor during aging? Does Prpf8 protein in brains of 4-weeks old Prpf8Δ366/+ drop so dramatically as in controls?

We did not observe any aging-associated neurodegeneration or other aging related phenotype in Prpf8Δ366/+ animals. We state that in the text (p. 14, middle of the page). We provide evidence that 12 week old animals do not show any difference in Prpf8 expression between Prpf8+/+ and Prpf8Δ366/+ animals (Fig. 8B,C). We now also include western blot from 4 week old animals (new Fig. S15B).

The observed postnatal regulation of U5 snRNP proteins, including Prpf8, but not their mRNAs is exciting. It is intriguing how mouse cerebellum/retina can function with about 15 % or lower levels of the original amount of Prpf8 protein from a few weeks postnatally. A similar regulatory phenomenon has been observed by Stankovic et al 2020. The authors noted that despite overexpressing wild-type or RP-Prp8 mutant variants in the developing Drosophila eye using a strong ey-Gal4 driver, the total amount of Prp8 protein did not exceed that of control, implying a regulation at the protein rather than mRNA level.

Thank you for pointing this out! It goes with the model that splicing factor abundance is, at least in some tissues, regulated at the protein and not mRNA level. We discuss and reference Stankovic et al . 2020 in Discussion (p. 17, middle paragraph).

March 15, 2023

RE: Life Science Alliance Manuscript #LSA-2022-01855R

Dr. David Stanek
Czech Academy of Sciences, Institute of Molecular Genetics
Department of RNA Biology
Videnska 1083
Prague 14220
Czech Republic

Dear Dr. Stanek,

Thank you for submitting your revised manuscript entitled "Retinitis pigmentosa associated mutations in Prpf8 cause degeneration of cerebellar granule cells". We would be happy to publish your paper in Life Science Alliance pending final revisions necessary to meet our formatting guidelines.

- please address the remaining Reviewer's 3 points
- please add the video legend to the main manuscript text
- please add the Twitter handle of your host institute/organization as well as your own or/and one of the authors in our system
- please add a figure callout for Figure S10 D-E and double-check your figure callouts for Figure S12-you have a callout for Panel A, but this panel isn't in the figure or legend; since it's the only panel in the figure, we do not need it designated with a letter

Figure Check:

-Figure S1C, 2nd row, looks like a splice before last 3 blots; Figure S1C (2nd blot section): looks like a splice in 2nd row in the middle and bottom row at the very beginning, there are inconsistencies. Please provide source data for these two figure panels.

A. FINAL FILES:

B. MANUSCRIPT ORGANIZATION AND FORMATTING:

Sincerely,

Reviewer #1 (Comments to the Authors (Required)):

In revising the manuscript, the authors have adequately taken care of the points raised by this reviewer.

Reviewer #2 (Comments to the Authors (Required)):

The revised version answered all my previous concerns and comments. A great manuscript providing a significant advance in the field of research. For me, the cerebellar phenotype is surprising. With these mice and findings, it will be much easier to continue the research on how Prpf8 is acting.

In the revised version, the RRIDs are still missing, but the research reagents are well documented - so - all fine. Thank you for this contribution.

Reviewer #3 (Comments to the Authors (Required)):

The authors provided satisfactory answers to most of my concerns. However, I would like to return to my first question, which seemed misunderstood and thus not addressed. Rather than asking for different types of splicing alterations (intron retention, alternative splice site usage etc.) I asked for changes in transcript biotypes other than protein-coding genes and circRNAs, such as transposable elements, lncRNAs, snoRNAs, etc., in control vs. Prpf8 mutant samples. The second part of the question was directed toward the specificity of circRNA dysregulation in response to Prpf8 mutations. Is the impact on circRNAs specific to the presence of Prpf8 mutations, or would changes in circRNAs, mainly the most abundant such as Rims2 circ_0000595, be observed in conditions where gene expression is deregulated to a similar extent by other spliceosome-independent insults?

In response to reviewers' comments, the authors modified the Discussion to reflect on the cell type specificity and the requirement for homozygosity to manifest the Prpf8 mutant phenotypes. The new text shall be corrected for typos and spelling errors.

Examples:

"Surprisingly, the other cell (delete) cerebral cells (e.g. Purkynje cells) were not affected and were able to survive despite

(should be "despite") significant granule cell decay, at least during the experiment."

"And why we had to breed mice to homogeneity (should be "homozygosity") when in humans the disease is autonomously dominant?"

In contrast, negative phenotype in RPE was observed in Prpf31Ala216Pro/wt rodents (Valdes-Sanchez et al, 2019). Ambiguous expression: Does "negative" mean aberrant or no phenotype?

In mice, we found higher levels of splicing proteins in retina than in cerebellum, which might explain, why granule cells, which represent the majority of cerebral cells, are primary (should be "primarily") targeted in the mouse model.

Response to reviewer #3

The authors provided satisfactory answers to most of my concerns. However, I would like to return to my first question, which seemed misunderstood and thus not addressed. Rather than asking for different types of splicing alterations (intron retention, alternative splice site usage etc.) I asked for changes in transcript biotypes other than protein-coding genes and circRNAs, such as transposable elements, lincRNAs, snoRNAs, etc., in control vs. Prpf8 mutant samples.

Thank you for clarifying the question. We analyzed various RNA biotypes and these data are presented in Supplementary table S2. To summarize our findings, we did not identify any non-protein coding transcripts that would be deregulated in both mouse strains in 4 week animals. In 8/12 week animals, we identified 20 lincRNAs that are misexpressed in the same direction in both Prpf8 N/N and Prpf8 delta17/delta17 strains with respect to control animals. We added a sentence to state this result in p. 8 - middle paragraph.

The second part of the question was directed toward the specificity of circRNA dysregulation in response to Prpf8 mutations. Is the impact on circRNAs specific to the presence of Prpf8 mutations, or would changes in circRNAs, mainly the most abundant such as Rims2 circ_0000595, be observed in conditions where gene expression is deregulated to a similar extent by other spliceosome-independent insults?

We did not test any other conditions that would affect spliceosome function and thus we cannot answer whether Rims1/2 (and other highly abundant) circRNAs change their expression upon splicing deregulation. However, it has been observed that inhibition of the spliceosome (e.g. by Isoginkgetin or downregulation of spliceosomal proteins including Prp8) enhance circRNA expression with respect to host mRNA. We cite these findings in our manuscript (p. 17 - bottom of the page).

In response to reviewers' comments, the authors modified the Discussion to reflect on the cell type specificity and the requirement for homozygosity to manifest the Prpf8 mutant phenotypes. The new text shall be corrected for typos and spelling errors.

We corrected all the typos and errors. Thank you!

March 21, 2023

RE: Life Science Alliance Manuscript #LSA-2022-01855RR

Dr. David Stanek
Czech Academy of Sciences, Institute of Molecular Genetics
Department of RNA Biology
Videnska 1083
Prague 14220
Czech Republic

Dear Dr. Stanek,

Thank you for submitting your Research Article entitled "Retinitis pigmentosa associated mutations in Prpf8 cause degeneration of cerebellar granule cells". It is a pleasure to let you know that your manuscript is now accepted for publication in Life Science Alliance. Congratulations on this interesting work.

DISTRIBUTION OF MATERIALS:

Again, congratulations on a very nice paper. I hope you found the review process to be constructive and are pleased with how the manuscript was handled editorially. We look forward to future exciting submissions from your lab.

Sincerely,
